# Programmable synthetic cell networks regulated by tuneable reaction rates

Adrian Zambrano[1,6], Giorgio Fracasso[1,2,6], Mengfei Gao [1,2,6], Martina Ugrinic[3], Dishi Wang[4,5], Dietmar Appelhans [4], Andrew deMello[3] & T-Y. Dora Tang [1,2✉]

Coupled compartmentalised information processing and communication via molecular diffusion underpin network based population dynamics as observed in biological systems. Understanding how both compartmentalisation and communication can regulate information processes is key to rational design and control of compartmentalised reaction networks. Here, we integrate PEN DNA reactions into semi-permeable proteinosomes and characterise the effect of compartmentalisation on autocatalytic PEN DNA reactions. We observe unique behaviours in the compartmentalised systems which are not accessible under bulk conditions; for example, rates of reaction increase by an order of magnitude and reaction kinetics are more readily tuneable by enzyme concentrations in proteinosomes compared to buffer solution. We exploit these properties to regulate the reaction kinetics in two node compartmentalised reaction networks comprised of linear and autocatalytic reactions which we establish by bottom-up synthetic biology approaches.

[1] Max Planck Institute of Molecular Cell Biology and Genetics, Pfotenhauerstraße 108, 01307 Dresden, Germany. [2] Cluster of Excellence Physics of Life, Technische Universität Dresden, 01602 Dresden, Germany. [3] Department of Chemistry & Applied Biosciences, ETH Zurich, Vladimir Prelog Weg 1, 8093 Zurich, Switzerland. [4] Leibniz-Institut für Polymerforschung Dresden e.V., Hohe Straße 6, 01069 Dresden, Germany. [5] Technische Universität Dresden, 01069 Dresden, Germany. [6]These authors contributed equally: Adrian Zambrano, Giorgio Fracasso, Mengfei Gao. ✉email: tang@mpi-cbg.de

The spatial and temporal orchestration of sophisticated dynamical biological behaviours, such as regulation and morphogenesis, occurs across length scales from molecules, to cells to tissues. In a reductionist description, the information processing performed by networks of biochemical reactions connected via intra and intercellular communication is required for complex behaviours. The reactions are compartmentalised within cells and subjected to external cues from communication with neighbouring cells in the population, for example, by molecular diffusion[1]. To begin to deconvolute the underlying architecture of network-based population behaviours it is crucial to probe how combined and emergent features of compartmentalised information processing and intercellular communication can affect dynamical systems. In particular, it is important to determine how compartmentalisation will affect biochemical reaction rates in order to rationally design and build compartmentalised reaction networks.

Bottom-up synthetic biology has advanced the ability to design and build features of network-based population behaviour[2], where chemical reaction networks[3–8], compartments with tuneable properties[9–13] or communication pathways between cells have been reported[14–16]. In addition, it has been shown that chemical reaction networks based on inorganic reactions, genetic circuits and PEN DNA reactions can be compartmentalised within water-oil emulsions[17–19], liposomes[20], microfluidic channels[21–24], or localised onto nanoparticles[25] where compartmentalisation and localisation affect the biochemical reaction rates. Recent studies have shown that compartmentalised non-enzymatic DNA strand displacement reactions undergo communication between proteinosomes by the diffusion of DNA[26,27]. Despite this progress, the ability to combine compartmentalised information processing based on dynamical enzymatic reactions capable of activation, initiation and degradation, coupled to molecular communication has not yet been demonstrated. These properties are key to building out-of-equilibrium behaviours within synthetic networks.

The PEN DNA toolkit, which utilises the polymerase (P), exonuclease (E) and nickase (N) enzymes, is ideally suited for building out-of-equilibrium information processing networks with different topologies[6,28,29]. Depending on the sequence of the DNA template strand; autocatalytic, linear and inhibitory reactions can be programmed. These reactions can be interconnected to build programmable and sophisticated networks that include bistability. The combination of enzymes, polymerase and exonuclease permits growth and degradation processes which provide the cornerstone for driving an out-of-equilibria system. In these reactions, a DNA primer strand or substrate (S) binds to a template strand (T) based on DNA base pairing (Fig. 1a (1)). Once bound, Bst polymerase (P) extends S along T (Fig. 1a (2)). The extended DNA strand is cut by a nickase (Nb.Bsml) (N) (Fig. 1a (3)) and the DNA dissociates to leave two primer strands, the substrate (S) and product (S') (Fig. 1a (4)). Exonuclease (E) degrades DNA strands which are not protected by phosphothiorionate modifications. In this work, we encapsulate the PEN DNA reactions within proteinosomes which are micron sized membrane-bound compartments formed from covalently linked protein-polymer conjugates. In our design, encapsulation of the DNA template sequence imparts a unique biochemical identity on the compartment with a specific information processing reaction (Fig. 1b). Here, template DNA can be encapsulated at a high local concentration within the proteinosomes but at low overall concentration in the total dispersion. The PEN DNA reaction within the proteinosome is triggered by the diffusion of enzymes, DNA primers (substrates) into the proteinosomes whose membrane allows the efflux of these materials out of the proteinosomes. The products of the reaction can therefore diffuse into neighbouring proteinosomes whilst simultaneously being diluted in the dispersion. As a consequence, these information processing units can be coupled together to form reaction networks of different topologies linked by molecular diffusion.

Herein, based on our design principle, we demonstrate spatially localised PEN DNA reactions within proteinosomes. Microfluidics was used to generate mondisperse proteinosomes encapsulating a DNA template that could support designer PEN DNA reactions. In addition, this platform was used to characterise the reaction kinetics of the autocatalytic PEN DNA reaction. Due to compartmentalisation, we observe kinetic behaviours that are not accessible in buffer solutions. For example, we obtain the same autocatalytic reaction rates within the proteinosomes with an order of magnitude lower overall template concentration compared to buffer solution. We also showed that reaction rates, as a function of template concentration, were an order of magnitude greater than those observed in buffer solution. In addition, by using low concentrations of exonuclease we demonstrate the ability to completely turn off the autocatalytic reaction, a behaviour not achievable in a non-compartmentalised system under the same conditions. Overall, these behaviours may be attributed to the ability of the proteinosomes to selectively confine the template DNA whilst permitting the diffusion and dilution of primer DNA from the reaction centre. Consequently, the platform is ideally suited for building reaction networks that combine the features of compartmentalised information processing and inter-compartment communication, as described previously. We demonstrate this by building simple two-node networks from PEN DNA reactions. Our work not only demonstrates a robust and reproducible approach to building compartmentalised reaction network system, this platform has allowed us to characterise the effect of spatial localisation coupled to passive diffusion on PEN DNA reaction rates. Our findings show that we can use bottom-up approaches for building synthetic systems with features of reaction-diffusion across networks, based on selective compartmentalisation and intercellular communication. These features are important for the rational design of compartmentalised reaction networks and therefore provide a framework for furthering our understanding of reaction dynamics in compartments. This is a fundamental property of biological systems and is crucial for the realisation of synthetic micro-compartments for applications in industrial and engineering applications.

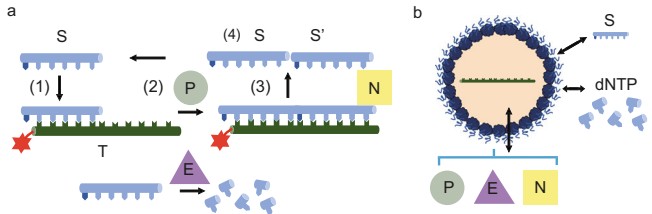

**Fig. 1 Design for compartmentalised PEN DNA reactions in proteinosomes. a** General description of PEN DNA reaction, (1) S binds to its complementary template T; (2) polymerisation by polymerase (P) of S along T, followed by nicking with nickase (N) to produce S and S'; (3) dissociation of S and S' from the template T. (4) Degradation of S and S' by exonuclease inside and outside the proteinosome maintains an out-of-equilibrium state for the system. **b** Cartoon of compartmentalised PEN DNA reactions. Template DNA (T) is encapsulated within the proteinosome. Substrate DNA (S); dNTPs; polymerase, exonuclease and nickase enzymes are loaded outside of the proteinosome and diffuse into the proteinosomes to initiate the reaction.

## Results

**Preparation and characterisation of DNA encapsulated proteinosomes.** First, we tested the ability to encapsulate phosphothiorionate modified template DNA within proteinosomes

using droplet-based microfluidics[30]. We expected to encapsulate the template DNA by size, therefore, fluorescently labelled template DNA tagged with biotin (7.8 kDa) was complexed to streptavidin (55 kDa) at a final molar ratio of 1:2 streptavidin:DNA (see "Methods") to generate a DNA-biotin-streptavidin complex of approximately 71 kDa. 1 μM of this complex was dissolved into an aqueous solution containing glucose oxidase (GOx) conjugate (4 mg mL$^{-1}$) (in 0.05 M HEPES, 0.01 M MgCl$_2$, 0.1 M of KCl, pH 7.6, See Supplementary Table 7) and flowed with an oil phase consisting of 2-ethyl-1-hexanol and BS(PEG)$_9$ (0.5–2 mM) at flow rates of 1.1 μL min$^{-1}$ and 2.5 μL min$^{-1}$ respectively. Water–oil droplets produced at the flow focus junction were collected and incubated at 4 °C for at least 12 h to ensure that all the protein-polymer conjugates had been cross-linked by BS(PEG)$_9$. The oil was then completely removed and replaced with Milli-Q water by step-wise dialysis with water and ethanol (see "Methods"). This produced water-in-water proteinosomes encapsulating DNA-biotin-streptavidin DNA complexes (see "Methods"). Using this methodology, it was possible to produce large populations of water-in-water proteinosomes (Supplementary Fig. 1). Size analysis of the proteinosomes using custom-made macro script in FIJI (Fiji is Just ImageJ) (see "Methods") showed a significantly smaller polydispersity index for droplets prepared by microfluidics (radius = 12.1 ± 1.8 μm, RSD = 14%) compared to those produced by bulk methodologies (radius = 24.0 ± 11.4 μm, RSD = 48%) (Supplementary Fig. 2). In addition, analysis of the fluorescence intensity of the DNA complex showed a normal distribution of DNA across the proteinosomes prepared by microfluidics compared to a log-normal distribution in the proteinosomes prepared by bulk methodologies. Our results show that microfluidics techniques are capable of producing proteinosomes with consistent size and template concentration, offering high levels of control and reducing variability within the population.

To determine the structure of the proteinosome containing the template DNA, proteinosomes were prepared with fluorescein isothiocynate (FITC) labelled protein conjugate and with carboxy-X-rhodamine (ROX) labelled DNA-biotin-streptavidin complexes. The fluorescently labelled protein conjugate was mixed in a 1:2 ratio of fluorescent conjugate : non-fluorescent protein conjugate. Dual colour confocal fluorescence microscopy showed proteinosomes that exhibited greater fluorescence intensity at the outer surface for both the DNA complex (Fig. 2bii) and the protein conjugate (Fig. 2biii). Photobleaching a cross-section of the proteinosomes showed no fluorescence recovery of the fluorescently labelled DNA or protein conjugate within the proteinosomes or on the outer edge of the proteinosome (Fig. 2c). Indicating that the DNA complex was isolated within the proteinosomes by chemical cross-linking to the protein conjugate rather than size exclusion as initially expected.

Our design principle for compartmentalised PEN DNA reactions is reliant on DNA primer and PEN enzymes diffusing into the proteinosomes from the exterior. Therefore, to confirm the diffusivity of these reactants, DY530 labelled 20-mer DNA or FITC labelled polymerase were loaded into a dispersion of proteinosomes containing DNA-biotin-streptavidin complexes (Fig. 2d). Fluorescence microscopy showed that the template DNA and polymerase diffuse into the droplet as the proteinosome showed fluoresence from polymerase and DNA within its interior (Fig. 2di, ii). Subsequent repeated whole droplet bleaching of the proteinosome shows that the DNA exchanges with the environment as the fluorescence intensity is completely refreshed in the droplet within seconds (Fig. 2e). In comparison, incomplete recovery of fluorescence recovery after photobleaching (FRAP) of FITC polymerase indicates that there is an irreversible interaction between polymerase and the proteinosomes. These results show that the membrane of the proteinosomes are porous and the enzymes and DNA diffuse from the outside of the proteinosome into the interior. Line profiles of the fluorescence intensity of template DNA showed an even distribution of DNA template within the proteinosome and in the outer aqueous phase (Supplementary Fig. 3). Further, FRAP analysis of the 20-mer DNA shows that the diffusion coefficient of the DNA within the proteinosomes (Supplementary Fig. 4) and in buffer solution (Supplementary Fig. 4) are the same within error (8.0 μm$^2$ s$^{-1}$ ± 1.4 μm$^2$ s$^{-1}$ and 7.1 μm$^2$ s$^{-1}$ ± 1.0 μm$^2$ s$^{-1}$ respectively) indicating that the diffusion of DNA within the proteinosomes is not hindered by the crosslinked protein conjugates. In comparison, there is an increase in fluorescence intensity from tagged polymerase within the proteinosome compared to the outer solution (Fig. 2dii). Assuming that there is no change in the quantum efficiency of FITC within the proteinosome the degree of sequestration (fluorescence inside the droplet/outside of the droplet) was estimated to be 5.7 ± 0.5. This increase in polymerase within the proteinosome could be attributed to weak interactions of the polymerase with the protein conjugate. FRAP analysis shows that the polymerase was mobile within the proteinosome, with approximately 30% immobile fraction and a diffusion coefficient of 5.7 ± 0.7 μm$^2$ s$^{-1}$ which is on the same order of magnitude as the DNA. These results indicate that a fraction of the polymerase interacts strongly whilst another fraction has weak and transient interations with the protein conjugate. As the molecular weight of exonuclease (73 kDa) and nickase (78 kDa) are very similar or less than the polymerase we assume that these enzymes would also have the capability to diffuse into the droplet. Indeed, determination of the partition coefficient (K) of a range of proteins from monomeric GFP (MW = 25 kDa, $K = 0.99 ± 0.17$), BSA (MW = 67 kDa, $K = 0.78 ± 0.17$), ttRecJ exonuclease (MW = 73 kDa, $K = 0.87 ± 0.06$) and alchohol dehydrogenase (MW = 141 kDa, $K = 1.01 ± 0.17$) within proteinsomes loaded with AF594-T$_2$-Biotin gave partition coefficients greater than 0.71 for all proteins (See Supplementary Methods, Supplementry Fig. 5 and supplementary Table 8). Our results also show that the DNA sequence or it's fluorescence label does not affect the partition coefficient, within error, indicating that all molecules will sequester to the same degree regardless of the molecules encapsulated within the proteinosomes. Taken together, our results show that template DNA is isolated and fixed within the proteinosomes and that the enzymes and primer DNA can freely diffuse in and out of the proteinosomes. In addition, the proteinosomes are permeable to proteins up to 141 kDa regardless of the sequence or the fluorophore encapsulated into the proteinosome.

**PEN DNA autocatalytic reactions in proteinosomes**. Next, we used fluorescence microscopy to determine whether the proteinosomes could support the PEN DNA reactions and that cross-linking the DNA complex to the proteinosomes does not stop the PEN DNA reaction (Fig. 1a). To this end, proteinosomes containing DNA template coding for an autocatalytic reaction, prepared by microfluidics were loaded into a glass capillary channel. The DNA content within the proteinosome, determined by calibration (see Supplementary Methods and supplementary Fig. 6), was 0.12 μM ± 0.02 μM from a total starting concentration of 1 μM of DNA complex. The reaction was triggered by the addition of a reaction buffer (Supplementary Table 4) containing DNA primer (substrate-S) (1 nM), polymerase (12.8 U mL$^{-1}$), nickase (400 U mL$^{-1}$), dNTPs (0.1 mM) and all the required salts (Supplementary Table 2) (see "Methods"). The capillary channel was then permanently sealed and fluorescence microscopy images were taken every 3 min for at least 180 min (see "Methods"). Fluorescent optical images showed an increase in EvaGreen fluorescence intensity within the proteinosomes over time (Fig. 3c). EvaGreen dye binds to the dynamically formed double-

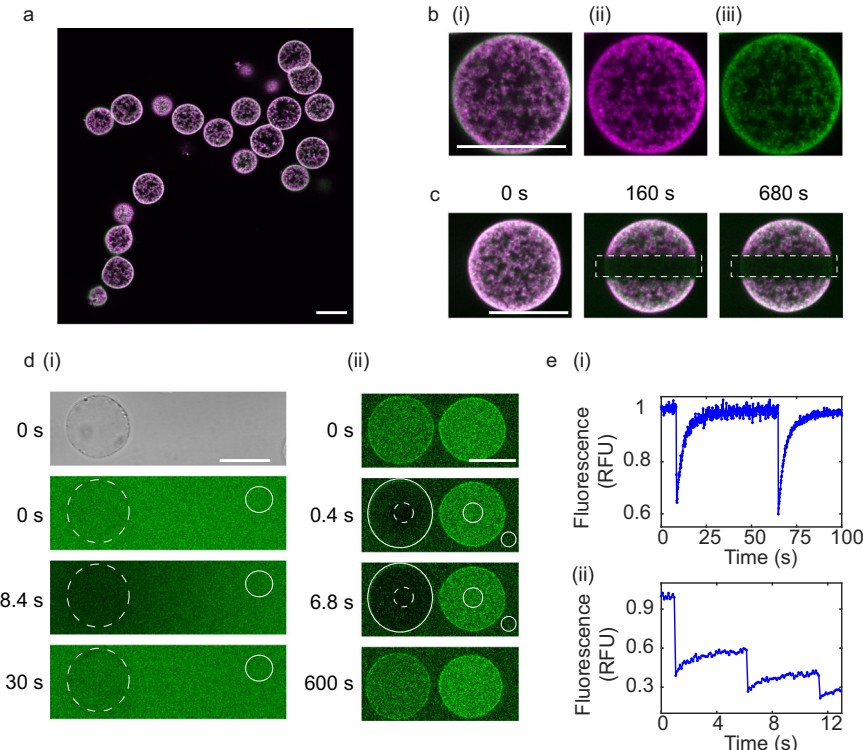

**Fig. 2 Fluorescence confocal microscopy of proteinosomes. a** Dual colour fluorescence confocal images of proteinosomes containing template DNA (protein conjugate labelled with fluorescein isothiocyanate (FITC) and DNA complex labelled with carboxy-X-rhodamine (ROX), **b** Zoom in of single proteinosome from: (i) Dual colour fluorescence, (ii) Single colour fluorescence channel, DNA labelled with ROX, (iii) protein conjugate labelled with FITC. All scale bars are 25 μm. **c** Fluorescent confocal microscopy images of photobleaching experiments, output frames for dual fluorescence confocal images of DNA-biotin-streptavidin complex labelled with ROX and protein conjugate labelled with FITC at $t = 0$ s (before bleaching), $t = 160$ s (after bleaching) and $t = 680$ s (after bleaching). Dotted rectangles show the region of photobleaching. All scale bars are 25 μm. Data are representative from at least 10 repeats. **d** Characterisation of the permeability of DNA (i) and PEN enzymes (ii) into proteinosomes. **d** (i) Bright-field image ($t = 0$ s) used to locate proteinosomes at $t = 0$ s. Output frames from Fluorescence Recovery after Photobleaching (FRAP) experiments of (i) single-stranded DNA (labelled with DY530) at $t = 0$ s (before photobleaching), $t = 8.4$ s (immediately after photobleaching) and $t = 30$ s (after recovery). Representation from two repeats. Scale bars are 20 μm. (ii) Polymerase enzyme tagged with FITC. Output frames at $t = 0$ s (before photobleaching), $t = 0.4$ s and $t = 6.8$ s and $t = 600$ s (after recovery). Scale bar is 20 μm. Dotted circle indicates bleached spot and the non-dotted circles show backgrounds used for normalisation from a single experiment. **e** FRAP analysis of (i) single-stranded DNA (ii) polymerase. Source data are provided in the source data file.

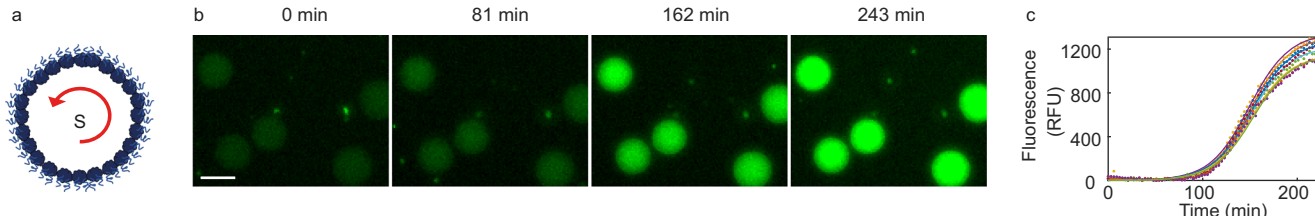

**Fig. 3 Proteinosomes contain and support autocatalytic PEN DNA reactions. a** Autocatalytic reaction, when $S = S'$ the PEN DNA reaction is autocatalytic. **b** Widefield optical fluorescence microscopy images show increasing EvaGreen fluorescence from DNA production within proteinosomes. Final total concentration of DNA template is 0.6 nM, initial concentration of primer/substrate is 1 nM at 42 °C. Images are a representation from a single experiment. Scale bar is 20 μm. **c** Autocatalytic growth curves obtained from image analysis show fluorescence increase from six individual proteinosomes. Source data are available in the source data file.

stranded DNA in a non-sequence-specific manner (Supplementary Fig. 7). In this experiment, this happens when the substrate strand is extended by the polymerase, or when the substrate or product strand binds to the template strand. Using custom-written FIJI analysis (Supplementary Fig. 8), the fluorescence intensity as a function of time was obtained from individual proteinosomes, showing characteristic autocatalytic profiles (Fig. 3c).

To further confirm that the PEN DNA reaction was completely isolated within the proteinosomes, 7.5 μL of the aqueous phase

was removed from the top of a dispersion of proteinosomes which had been allowed to completely sediment. The aqueous phase was incubated with the PEN enzymes, the reaction buffer, EvaGreen dye and either 0.1 mM dNTPs or without dNTPs to determine whether any reaction would take place outside of the proteinosomes. After incubation at 42 °C, spectroscopic measurements showed no increase in fluorescence intensity from the PEN DNA reaction. In comparison, experiments undertaken with a resuspended dispersion of proteinosomes showed an increase in fluorescence intensity from increasing DNA concentration

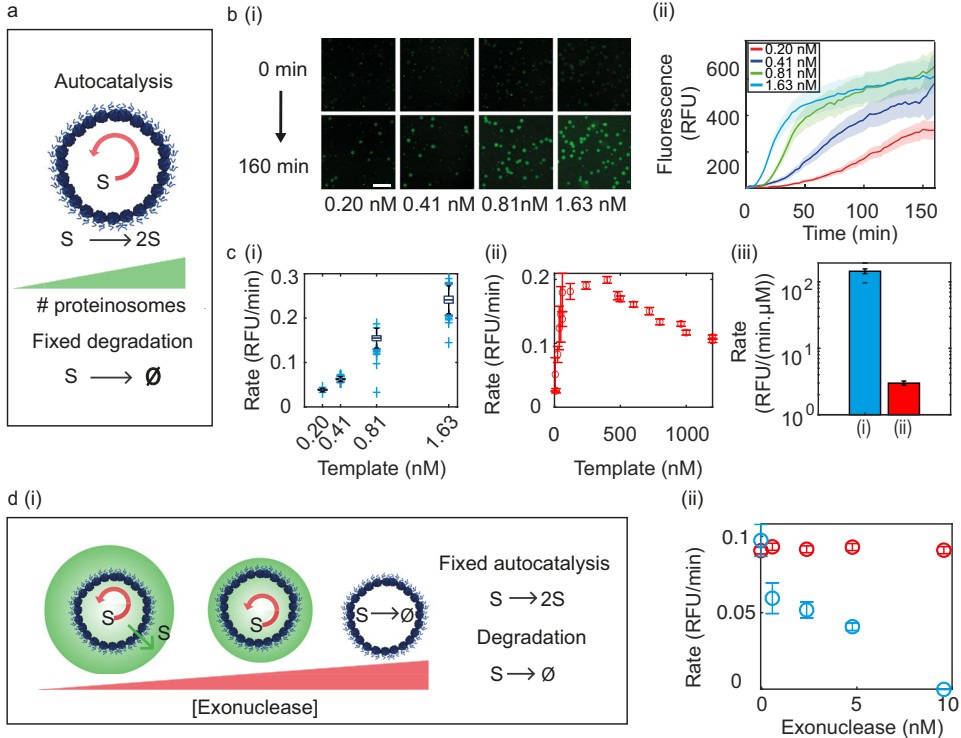

**Fig. 4 Characterisation of autocatalytic PEN DNA reactions within proteinosomes. a** Increasing overall template concentration at fixed exonuclease concentration (0.6 nM). **b** (i) Fluorescence widefield microscopy images, at $t = 0$ min and 160 min, showing proteinosome populations at different densities resulting in total DNA template concentration of 0.20–1.63 nM. Images are a representation from a single experiment. Scale bar is 100 μm. (ii) Autocatalytic kinetic curves from microscopy images. Bold line is the average kinetic profile from 54, 67, 99 and 436 proteinosomes for total DNA concentrations of 0.20 (red), 0.41 (blue), 0.81 (green), 1.63 nM (cyan) respectively. Shaded region is the standard deviation. **c** Reaction rates for autocatalytic reactions carried out with varying template concentration in (i) proteinosomes, box plots show the the median, first and third quartile and upper and lower adjacent values. The blue data points are the outside points. Plots are from 107, 162, 370 and 669 proteinosomes. (ii) In buffer solution. The initial rates were obtained from fitting of the kinetic profiles and plot as a function of template concentration. Error bars represent the error from the fit. (iii) Bar chart shows the linear fit of autocatalytic reaction rate per template ($dr/d[T]$) for (i) proteinosomes (144.5 ± 44.0 RFU min$^{-1}$ μM$^{-1}$) (blue), (ii) buffer (1.6 ± 0.3 RFU min$^{-1}$ μM$^{-1}$) (red). Error is obtained from the fit to the data. **d** (i) with constant template concentration and increasing exonuclease concentration, the exonuclease degrades the substrate inside and outside of the proteinosomes. With high exonuclease concentration, the reaction will be completely switched off. (ii) Initial rates of autocatalytic reaction with increasing exonuclease concentration in proteinosomes (blue squares) and a buffer solution (red circles). Rates and standard deviations were obtained from fits to the kinetic profiles. Data obtained from one experiment. Source data are provided in the source data file.

produced by the PEN DNA reaction (Supplementary Fig. 9). Taken together our results show that the PEN DNA reaction is taking place within the proteinosomes and any reaction taking place outside of the proteinosomes is negligible. In addition, our results show that the chemical modifications required to produce the proteinosomes does not inhibit the PEN DNA reaction.

**Effect of compartmentalisation on PEN DNA autocatalytic reactions**. As the proteinosomes have the ability to retain high concentrations of template DNA whilst allowing the influx and outflux of primer DNA their reaction kinetics may not be analogous to reaction rates within buffer solution. We, therefore, characterised the effect of varying the total template concentration and exonuclease concentration on the rate of the autocatalytic reaction (Fig. 4) in buffer solution and when the reaction was localised within the proteinosomes.

As the concentration of template DNA within the proteinosome was fixed at 0.12 μM, the total concentration of DNA template in the dispersion was varied by altering the density of proteinosomes and was typically between 0.2 and 1.63 nM of DNA (Fig. 4b). The total concentration of template DNA in the dispersion was determined by measuring the total volume of proteinosomes within the capillary channel and obtaining the

total concentration based on the assumption that each proteinosome contains 0.12 μM of DNA (supplementary Figs. 6 and 10). The PEN DNA reaction was triggered by the addition of reaction buffer as described previously and the fluorescence intensity in each individual proteinosome was obtained as a function of time for each total DNA concentration (Fig. 4b). In order to make quantitative comparisons between autocatalytic reaction profiles, under different reaction conditions, we used an exponential fit based on a logistic model to the initial time points which account for the acceleration and deceleration of the reaction to obtain the rate (see "Methods"). In order to make semi-quantitative comparisons between our experiments, we undertook a numerical differentiation of the data to obtain the discrete differences of fluorescence signals between every time point. This was fit to a Gaussian and used to determine the mid-point to decay ($t = f$) (Supplementary Fig. 11). The exponential fit was then applied from $t = 0$ to $t = f$ to obtain the rate of reaction. In cases where it was not possible to determine the mid-point to decay using this method, the data were fit to a defined time point $t = f$ which provided statistically sufficient data points for analysis. Using this approach we obtained the rates of autocatalytic reaction for increasing template concentration in a buffer and within proteinosomes.

For the buffer solution, we observed an increase in the autocatalytic rate between 0 and 0.4 μM DNA template to a maximum rate of $0.20 \pm 0.01$ RFU min$^{-1}$ (Fig. 4ci, ii, Supplementary Fig. 12). Further increases in DNA template concentration led to a decrease in the initial rate. This observation may be attributed to product inhibition, where large concentrations of the substrate strand will inhibit the autocatalytic reaction. Primer DNA can bind to the template strand at the 5′ end and the 3′ end. As more primer is produced in the autocatalytic reaction, the probability to bind to both sites on the template strand will increase leading to product inhibition and a reduction in the overall rate.

Analysis of the autocatalytic rates within proteinosomes (Fig. 4c) shows that the same rates of reaction are achievable within the proteinosome compared to the buffer solution with an order of magnitude lower total template concentration. Further analysis of the rates of reaction as a function of template concentration (d$r$/d$[T]$) (RFU min$^{-1}$ μM$^{-1}$) (Fig. 4ciii, Supplementary Fig. 13) shows that d$r$/d$[T]$ is over an order of magnitude greater in the proteinosomes ($144.5 \pm 44.0$ RFU min$^{-1}$ μM$^{-1}$) compared to the buffer ($1.6 \pm 0.3$ RFU min$^{-1}$ μM$^{-1}$).

Such a dramatic increase in the rate of reaction could be attributed to the (six times) increase of polymerase within the proteinosomes as shown. To check for this, we undertook control experiments to determine whether increasing the polymerase concentration could account for the observed increase in the rates of reaction with increasing template concentration. In buffer solution, our results showed (Supplementary Fig. 14) less than a two-times increase in the autocatalytic rate with a sixfold increase in the polymerase concentration, $r_{12.8 \, \text{U mL}^{-1}} = 0.092 \pm 0.004$ min$^{-1}$ and $r_{73 \, \text{U mL}^{-1}} = 0.16 \pm 0.01$ min$^{-1}$ indicating that a six times increase in polymerase concentration could not account for the dramatic increase in the reaction rate within the proteinosomes compared to the buffer. As the FRAP data shows that the polymerase interacts with the proteinosome, consequent alterations to the polymerase activity can not be entirely ruled out. An additional possible explanation for the increased rate of reaction as a function of template concentration could be attributed to the porosity of the proteinosome and the ability for the substrate/product DNA to diffuse from the reaction site once it has been produced. As a consequence, the local concentration of the primer at the reaction site remains low which reduces the probability for the primer to bind to both the 3′ and 5′ end of the template strand thus relieving the effects of product inhibition within the compartment.

If this were the case, and the primer diffusing out of the proteinosome is diluted within the dispersion, it should be true that by increasing the exonuclease concentration and therefore the rate of degradation inside and out of the proteinosomes the autocatalytic rates will decrease. To this end, we characterised the rates of reaction with increasing exonuclease concentration (Fig. 4di, Supplementary Fig. 15). The results show that in a dispersion of proteinosomes ([DNA template]$_{\text{total}} = 0.06$ nM) and increasing exonuclease concentration the autocatalytic rates decrease until the reaction is completely switched off at 9.6 nM exonuclease. In comparison, increasing the exonuclease concentration in buffer solution, with 1 μM of template, showed an initial decrease in the rate after the addition of 0.6 nM of the exonuclease. Upon further increase of exonuclease concentration (up to 9.6 nM), the rates remained constant at approximately 0.1 min$^{-1}$ (Fig. 4dii). The results indicate that changing the exonuclease concentration in the buffer solution by ten times does not lead to any change in the rate and is commensurate with a DNA production rate that is greater than the rate of degradation. Whilst, in the compartmentalised system, there is a fine balance between the rate of degradation and the rate of production in the proteinosome system. Compartmentalisation of

the template DNA enables a high local concentration of template at 0.12 μM but a low total concentration (nM) in the total dispersion. Consequently, total primer production is on the same order of magnitude as exonuclease degradation. This leads to stronger handles in tuning the reaction rates by small changes in exonuclease concentration within a proteinosome dispersion compared to the buffer solution.

Taken together, our results show that the same rates of reaction are accessible in buffer and proteinosomes with an order of maginitude difference in the total template concentrations in the proteinosomes (nM) compared to the buffer (μM). One explanation for this observation could be that the local template concentration in the porous proteinosome is high and the DNA produced from the reaction can diffuse away from the reaction centre. The rate of reaction as a function of template concentration is increased by an order of magnitude compared to buffer as local build-up of primer is prevented thus relieving product inhibition. Moreover, as the overall concentration of template is on the nM regime, the overall primer production is also low and the degradation can be tuned to be on the same order of magnitude as the production. This offers a controllable handle to switch off the autocatalytic reactions—a behaviour which is not observed in buffer solutions under similar conditions. Our results show that spatial compartmentalisation with free diffusion of substrates and products offers several kinetic advantages compared to undertaking the reaction in buffer. This includes increased and tuneable reaction rates based on varying the template and enzyme concentrations. Overall, the results show that compartmentalisation plays a strong role in altering reaction landscapes indicating that reaction networks based on PEN DNA will have different behaviours in buffer.

## Two-node networks based on compartmentalised PEN DNA reactions and intercellular communication.

Building on our findings and exploiting the ability for the proteinosome to spatially localise the template but permit diffusion of the primer/substrate DNA through the permeable membrane. We next tried to increase the complexity of the system by building a 2 node network based on two different compartmentalised PEN DNA reactions that are connected via intercellular molecular diffusion between proteinosomes. Here we used two different template strands to programme a linear reaction ($T_1$) and an autocatalytic reaction ($T_2$) where the product of the linear reaction is required to trigger the autocatalytic reaction (Fig. 5a, b). The two-node reaction cascade was prepared by mixing two populations of proteinosomes containing DNA complexes comprised of either a fluorescently labelled template strand coding for a linear reaction ($T_1$-ROX) or a fluorescently labelled template strand coding for the autocatalytic reaction ($T_2$-DY530) in a capillary channel. Fluorescent optical microscopy images showed two different populations of proteinosomes (Fig. 5ci). The reaction was triggered by the addition of a reaction buffer as described previously (Supplementary Table 3).

Fluorescent optical images showed that there was an increase in EvaGreen fluorescence intensity as time progresses in each of the droplet populations (Fig. 5c). Analysis of the fluorescence increase in the two populations showed an increase followed by a decrease in fluorescence intensity for population 1 and a delayed onset of an autocatalytic reaction in population 2 (Fig. 5d). Control experiments for the two-node network which involved incubating proteinosomes containing the template for the autocatalytic reaction ($T_2$) only with the primer strand for the linear reaction ($S_1$) showed no increase in EvaGreen fluorescence over time (Fig. 5e, Supplementary fig. 16). Our results indicate that the chemical reactions based on PEN DNA reactions can be localised within the proteinosomes and interconnected together by primer communication to build chemical reaction networks.

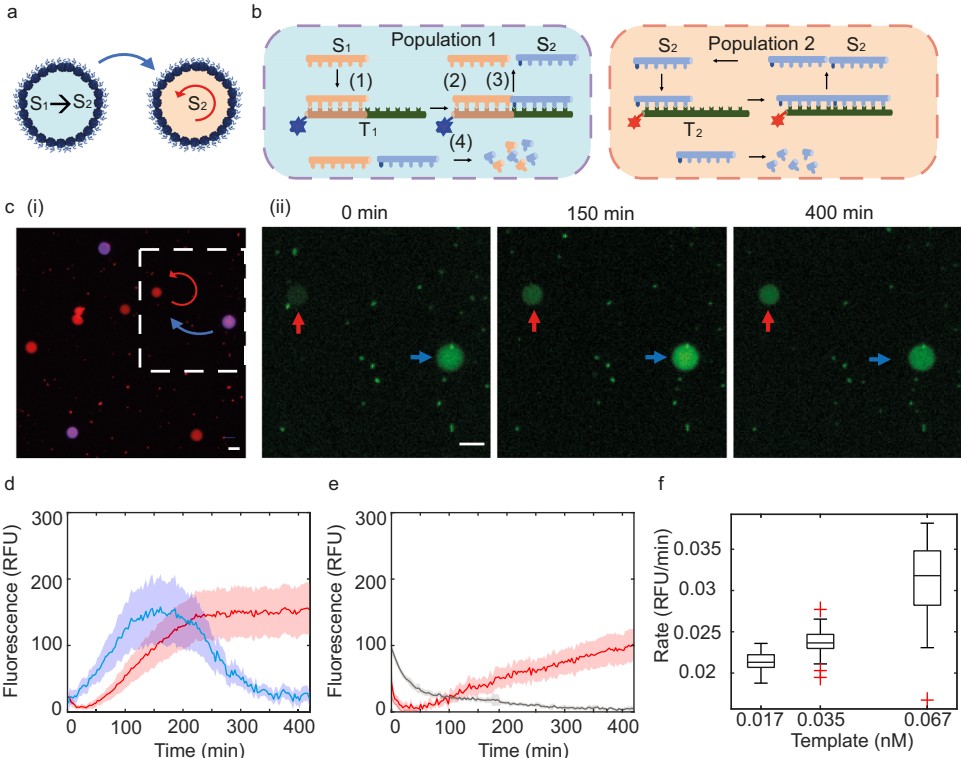

**Fig. 5 Two-node networks based on encapsulated PEN DNA reactions within proteinosomes. a** The product of a linear reaction programmed into population 1 activates autocatalysis in population 2. **b** Two different populations of proteinosomes are produced: population 1 contains ROX labelled $T_1$ and population 2 contains DY530 labelled $T_2$. **c** (i) Dual-channel widefield microscopy show two populations of proteinosomes. Population 1 (purple/blue) containing DNA template $T_1$ labelled with ROX and population 2 (red) containing DNA template $T_2$ labelled with DY530. (ii) Time-lapse widefield fluorescence microscopy images show increased fluorescence intensity from non-sequence-specific intercalation of Evagreen Dye due to DNA production inside the proteinosomes for populations 1 (blue arrow) and 2 (red arrow). Images are representations from two replicate experiments. Scale bar is 20 μm. **d** Kinetic fluorescence profiles obtained from the average kinetic profiles (bold line) and standard deviation (shaded region) of individual proteinosomes from populations 1 ($n = 13$) (blue curve) and 2 (red) ($n = 30$) at a $T_1$ and $T_2$ template ratio of 1:1 (0.067 nM: 0.068 nM). **e** Control experiment for the two-node network population containing population 2 proteinosomes (final $T_2$ concentration of 0.068 nM), $S_1$ primer (50 nM) and in the absence of $T_1$ proteinosomes. The bold red curve is obtained from the average kinetic profiles of $T_2$ proteinosomes ($n = 8$). The grey curve is obtained from the average kinetics of the background (9 circular regions with diameter of 20 pixels in the background ($n = 9$)). The shaded colours in the plots in a-c are the standard deviations from the kinetic curves from individual proteinosomes in their corresponding populations. **f** Reaction kinetics of a two-node network cell population. Box plot showing the initial rates of the autocatalytic reaction (population 2) as a function of template concentration ($T_1$) for 0.017 nM ($n = 23$), 0.035 nM ($n = 29$) and 0.067 nM ($n = 30$). The central line and yellow open circle represent the median line and the first and third quartile, error bars represent the upper and lower adjacent values and the blue data points are the outside points. Source data are provided in the source data file.

As control of the rates of reaction based on the primer concentration is an important parameter for designing and controlling reaction networks, we next varied the production of the primer DNA by the linear reaction to regulate the rate of the autocatalytic reaction. To do this, we varied the concentration of proteinosomes containing $T_1$. The concentration of population 2 was kept constant whilst the concentration of population 1 was varied (Fig. 5f, Supplementary fig. 17) at $T_1$:$T_2$ ratios of 1:1 (0.067 nM:0.068 nM) (Fig. 5d), 0.5:1 (0.035 nM:0.067 nM) (Supplementary Fig. 17), and 0.25:1 (0.017 nM: 0.068 nM) (Supplementary Fig. 17). Analysis of the autocatalytic rates from population 2 showed that the initial rate increases almost two times by increasing the proteinosome concentration in population 1 by a factor of 4 (Fig. 5f). Our results demonstrate the ability to spatially localise two different reactions which communicate with one another with autocatalytic reaction rates tuneable by the concentration of population 1. This not only demonstrates communication between proteinosomes but also addresses the fact that many reaction networks need delay nodes, which are mostly linear responses, to increase robustness. For example, negative feedback networks capable of oscillating can oscillate in a

larger region of parameter space with appropriate amounts of delay. Additionally, linear activation in a non-heterogenous, i.e. non-randomly, distribution of proteinosomes will be key to explore pattern formation.

## Discussion

In summary, we show that we can use a microfluidic technique to produce highly monodisperse populations of DNA-programmed proteinosomes. Our characterisation shows that template DNA is spatially localised within the proteinosomes where the DNA-biotin-strepatividin complex is chemically crosslinked to the protein conjugate and is immobile. The encapsulated DNA template is externally activated by the addition of PEN enzymes and substrate DNA which permeate into the centre of the proteinosomes. The reaction is programmed by the design of the template DNA. We have compared the rates of the autocatalytic reactions within the proteinosomes and in buffer under different conditions including, changing the template concentration and the exonuclease concentration. Autocatalytic rates as a function of template concentration are increased within the proteinosomes

compared to buffer at much lower DNA template. One possible explanation for this is that the ability for the proteinosome to localise template DNA at high concentration whilst permitting the diffusion and thus dilution of DNA product away from the reaction site. We show that varying the template concentration and exonuclease concentrations permits the ability to tune the rates of reaction. The characterisation of this property is vitally important for designing and building reaction networks based on localised PEN DNA reactions. One important aspect of designing and building robust networks is the controlled production of a substrate that triggers a second reaction as opposed to providing all the substrate at the beginning of the reaction. To this end, we have designed and built two-node networks based on programmed proteinosomes where a linear reaction population produces the substrate strand for an autocatalytic population. We further show that we can control the initial rate of autocatalysis in population 2 by varying the concentration of population 1.

Our study shows that we have developed a basic toolkit for building reaction cascades of localised chemical reactions based on PEN DNA reactions within proteinosomes. Where the proteinosomes encapsulate the template DNA code for specific reactions. Previous studies have shown the ability to programme networks based on DNA strand displacement within proteinosomes, however, the addition of enzymes will greatly facilitate the ability to drive the systems into an out-of-equilibrium state. Compartmentalisation also offers the ability to spatially localise, alter reaction kinetics and reduce the total amount of DNA template required for the reaction. Coupled with the coding ability of DNA, this platform allows access to a large combinatorial space. Our toolkit opens many possibilities for spatial multiplexing using a flexible and modular system, thus providing a general route for building synthetic compartmentalised reaction networks, based on reaction-diffusion mechanisms and a minimal number of components.

Whilst it is still a challenge to probe the effect of compartmentalisation in biological cellular systems, the ability to build micron-sized compartments encapsulating enzyme reactions has offered an unique opportunity to address this challenge without biological complexity. Our work represents an important step in bottom-up synthetic biology approaches by combining it with quantitative approaches. As a consequence we have shown that the kinetic landscape of compartmentalised reactions are not equivalent to free solution. Our results show that it is imperative, that biological behaviours based on networks of chemical reactions have to be considered within the context of compartmentalisation. Simplfied physical models of compartments are ideal for exploring these fundamental questions. Moreover, these minimal models could be effective as physical models of biological compartmentalisation where membrane-bound compartments which contain reactions allow the passive diffusion of molecules such as in mitochondria or in gram-negative bacteria.

## Methods

**Designing two-node PEN DNA reactions**. The template DNA for $S_1$ and $S_2$ has been described in previous publications[31]. The DNA substrate template and substrate for the linear activation reaction $S_1$ to $S_2$ was designed and checked using the open-source software NUPACK[32]. It was used to obtain the binding constant of the substrate DNA onto the template DNA which was then input into DACCAD[33] to simulate the connectivity of the two-node network consisting of a linear activation reaction and autocatalysis. Additionally, the substrate DNA was designed to inhibit any binding between it and the product. Six nucleic acid recognitions site for the nickase were designed so that five of the recognition sites were located on the substrate and one was located on the product. (See Supplementary Table 1).

**Biotinylated-DNA-streptavidin conjugation**. DNA-biotin (7.8 kDa) was conjugated to streptavidin to produce a DNA complex with increased molecular weight (71 kDa). To do this, 10 μL of biotinylated ssDNA ($T_2$ or $T_1$) (100 μM in MiliQ water) was added in 4 aliquots into 100 μL of Tris-EDTA, (10 mM Tris, 1 mM EDTA, pH 8) containing streptavidin (5 μM). The final concentration of

ssDNA was 10 μM and for streptavidin it was 5 μM, to achieve a final molar ratio of 1:2 Streptavin:DNA. After every 2.5 μL addition of DNA, the solution was vigorously shaken at 1000 rpm for 10 min at 25 °C. Frozen aliquots of 10 μM DNA-biotin-streptavidin solution were stored at −20 °C until use.

**Fluorescent labelling of Glucose oxidase (GOx) or polymerase**. Polymerase was dialysed against PBS buffer (137 mM NaCl, 2.7 mM KCl, 10 mM Na₂HPO₄, 1.8 mM KH₂PO₄, pH 7.4) to remove all of the storage buffer and then solvent exchanged with 0.1 M NaHCO₃ buffer, pH 8.4 using an Amicon filter (10 K) at 4 °C. The polymerase solution was adjusted to 5 mg mL⁻¹. Alternatively, 5 mg mL⁻¹of glucose oxidase was dissolved in 0.1 M NaHCO₃ buffer pH 8.4. Fluorescein isothiacyanate (FITC) was dissolved in anhydrous DMSO at a concentration of 1 mg mL⁻¹ then 50 μL of FITC was added dropwise for every mL of protein solution whilst stirring. The glucose oxidase or polymerase/FITC solution was incubated, with stirring, overnight at 4 °C in the dark. The solution was then dialysed against Milli-Q water using membrane spectra/POR 4 dialysis membrane MWCO 12–14 kDa, standard RC tubing. The concentration of the protein and FITC was determined by UV Vis at 280 nm and 490 nm respectively.

### Preparation of proteinosomes

*RAFT agent synthesis and polymerisation and characterisation of PNIPAAm.* The synthesis of bis(propylsulfanylthiocarbonyl) disulfide and PNIPAAm were performed according to literature methods[30,34]. The PNIPAAm was characterised by gel permeation chromatography. The dispersity ($Đ = M_w/M_n$; $M_n$ is the number average molecular weight; $M_w$ is the weight average molecular weight of block copolymers were detected using a size exclusion column equipped with a MALS detector (MiniDAWN-LS detector, Wyatt Technology, California, USA) and a viscosity/refractive index (RI) detector (ETA-2020, WGE Dr. Bures, Brandenburg, Germany). The column (PL MIXED-C with a pore size of 5 μm, 300 × 7.5 mm) and the pump (HPLC pump, Agilent 1200 series) were from Agilent Technologies (California, USA). 2% vol water in dimethylacetamide (DMAc) and 3 g L⁻¹ of LiCl were flowed at a rate of 0.5 mL min⁻¹ to elute the polymer. Pol(2-vinylpyridin) was used as a standard at 2 mg mL⁻¹ after filtration through a 0.2 μm filter. The data were processed using Cirrus GPC offline GPC/SEC software (version 2.0). The PNIPaam chain had an ($M_n$) of 23,000 g/mol, a molecular weight ($M_W$) of 27,000 g/mol resulting in a dispersity index ($Đ = M_w/M_n$) of 1.17 unless otherwise stated.

*Synthesis of protein-PNIPAAm conjugates.* Preparation of protein-PNIPAm conjugates was carried out as described previously[30]. Briefly, native glucose oxidase (GOx) or FITC labelled GOx was cationized, via carbodiimide chemistry, with aminohexane to increase the number of available amine groups. Subsequently, cationized glucose oxidase (GOx-NH₂ or GOx-FITC-NH₂) was reacted with mercaptothiazoline-activated PNIPAAm resulting in protein-PNIPAAm conjugates with a final molar ratio of 1:2 GOx-NH₂: PNIPAAm (determined by dry weight analysis as previously described) unless otherwise stated. The ratio of two polymer chains per protein was yielded by reacting these two reactants at an equal mass ratio in PBS or NaHCO₃ buffer (pH 8.4).

*Preparation of microfluidic devices.* Microfluidic devices were prepared as previously described in ref. [30]. Briefly, master moulds were fabricated using standard photolithography methods, and involved spin-coating two layers of SU-8 2010 photoresist (Microchem, USA) onto a silicon wafer (Silicon Materials, Germany) to yield a total height of 40 μm. All masters were treated with chlorotrimethylsilane (Sigma Aldrich, Switzerland) vapour in a vacuum desiccator to prevent adhesion of PDMS during moulding and demolding.

Microfluidic devices were fabricated by pouring a 4-mm-thick layer of polydimethylsiloxane (PDMS, Sylgard 184A: B, 10:1, Dow Corning) onto the appropriate master. The PDMS was cured for approximately 24 h at 70 °C, peeled off the master and Gauge 19 access holes punched. Devices for the production of proteinosomes were then bonded to a thin layer of PDMS after air plasma treatment. Lastly, all devices were bonded to a glass slide (24 × 75 mm, ThermoScientific, Switzerland). The assembled device was placed in an oven at 70 °C, overnight, to ensure complete bonding.

*Proteinosome preparation.* Proteinosomes were generated either via a microfluidic device[30] as previously described or via bulk methodologies to produce water-oil emulsions. For microfluidic methods, typically, the aqueous solution consisted of protein-PNIPAAm (4 mg mL⁻¹), DNA-biotin-streptavidin conjugates (1 μM DNA) was dissolved in 1× HEPES buffer at pH 7.6. The oil phase comprised of amine-reactive crosslinker BS(PEG)₉ (0.5–2 mM) in 2-ethyl-1-hexanol. Water–oil emulsions were generated using a flow focus PDMS microfluidic device as described previously[30] with flow rates of 2.5 μL min⁻¹ and 1.1 μL min⁻¹ for the oil and water phase respectively using Syringe pumps (neMesys Syring Pumps, Cetoni, Gmbh, Korbussen, Germany) and the software provided by the company.

Emulsions produced by bulk methodologies were achieved by shaking the vial containing a water/oil mixture, made up of 60 μL of aqueous phase in 1 mL of oil phase, ten times. The devices were loaded onto a Zeiss Axiovert 200 M inverted widefield microscope equipped with a 10× objective (Plan-Neofluar 10x NA 0.3

Ph1), a 16-channel CooLED pE-4000 (Andover, UK), an Andor Zyla PLUS sCMOS camera (Oxford instruments Belfast Northern Ireland) and a high speed PCO Dimax S4 monochrome sCMOS camera (Kelheim Germany).

The crosslinked water-oil emulsions produced by microfluidics or by hand were transferred to water by dialysis in 75 % ethanol/water followed by 35% ethanol/water and finally in Milli-Q Water. The water-in-water proteinosomes containing biotinylated-DNA-streptavidin were collected in a 2 mL Eppendorf tube, allowed to sediment overnight and concentrated by removing 1.5 mL of aqueous solution. The approximate concentration of the proteinosomes prepared by microfluidics were obtained by measuring the fluorescence of DY530 tagged to the DNA template using the SPARK 20 M (TECAN AG, Mannedorf, Switzerland) with wavelength/bandwidth $\lambda_{exc} = 537/7.5$ nm, $\lambda_{em} = 570/15$ nm. Confocal microscopy images of proteinosomes (in 1× HEPES buffer, pH 7.6, Supplementary table 7) encapsulating DY530 tagged DNA were analysed using FIJI macro scripts and fit using MATLAB and Python.

**Optical imaging of proteinosomes**. Water-in-water proteinosomes were imaged using a 10× objective (Plan-Neofluar 10× NA 0.3 Ph1, Zeiss) mounted onto a Zeiss Axiovert 200 M inverted widefield microscope equipped with the 16-channel CooLED pE-4000 and an Andor Zyla PLUS sCMOS camera in phase contrast mode or fluorescence. The microscope was equipped with the following filter sets: $\lambda_{ex} = 542/27$ nm and $\lambda_{em} = 593/46$ nm Beamsplitter H 560 LPXR for imaging DY530, $\lambda_{ex} = 575/15$ nm and $\lambda_{em} = 641/75$ nm Beamsplitter HC BS 596 for imaging carboxy-X-rhodamine (ROX), $\lambda_{ex} = 475/28$ nm and $\lambda_{em} = 525/15$ nm Beamsplitter H 488 LPXR for imaging EvaGreen.

Fluorescent confocal images were undertaken with objective Zeiss C-Apochromat 40× NA 1.2 W objective mounted on a confocal Zeiss LSM 880 Airy inverted microscope. Samples were loaded into capillary channels (ID 1.0 × 0.1 mm, 0.7 mm thickness) and then sealed with 5 min epoxy glue. DY530 labelled DNA was imaged with $\lambda_{exc} = 514$ and $\lambda_{em} = 544$–695 nm or for FITC labelled protein conjugate with $\lambda_{exc} = 488$ and $\lambda_{emis} = 499$–535 nm. Size analysis and distribution of DNA within individual proteinosomes containing fluorescently labelled DNA were analysed using a custom-written analysis macro script using FIJI (see SI and data availability statement).

**Image analysis from kinetic experiments**. Optical microscopy images were analysed using FIJI. To obtain reaction kinetics from optical microscopy experiments containing individual proteinosomes. A FIJI macro script (see data availability statement) was used to detect proteinosomes based on the fluorescence from the DY530 labelled DNA. The identified regions of high fluorescence were then blurred to avoid empty pixels during image thresholding and then binarized using the "watershed" plugin to separate proteinosomes in close proximity to each other. The "Analyze Particles" plugin with 0.7–1.0 circularity and 17-25 μm was used to detect and generate a list of images detected using the Evagreen fluorophore as a function of time.

**Determination of proteinosome concentration**. To determine the number of proteinosome in each experiment, a calibration curve relating the number of proteinosomes to their total intensity of fluorescence was obtained (Supplementary Fig. 10). To do this, proteinosomes containing 0.12 μM of DNA template, prepared by microfluidics, were serially diluted with PBS buffer. The fluorescence intensity for each dilution was obtained using the TECAN SPARK 20 M spectrometer (Wavelength/bandwidth: $\lambda_{ex} = 530/7.5$ nm and $\lambda_{em} = 570/15$ nm) and the number of proteinosomes were counted from widefield microscopy images obtained on the Zeiss Axiovert 200M inverted widefield using a 10× objective (Plan-Neofluar 10× NA 0.3 Ph1) and Andor Zyla PLUS sCMOS camera (Oxford instruments Belfast Northern Ireland) microscope using custom FIJI macro script (see data availability statement). Stocks of proteinosomes of known concentration were stored at 4 °C until the experiment. Concentrations of proteinosomes were determined by comparing the fluorescence intensity of a known volume of proteinosomes to the calibration curve. To correct for proteinosome number variability between experiments, the number of proteinosomes was always counted after the experiment using custom FIJI macro script.

**Fluorescence recovery after photobleaching (FRAP)**. FRAP was undertaken for proteinosomes containing the DNA complex. Proteinosomes were prepared by microfluidics with FITC labelled protein conjugate and ROX labelled biotin-streptavidin DNA from proteinosomes. The rate of diffusion of DY530 labelled 20-mer DNA and FITC labelled polymerase into the proteinosome was determined by whole droplet FRAP. To do this, dispersions of proteinosomes were incubated with either 10 μM of DY530 labelled DNA or 10.2 μM of FITC labelled polymerase to check for repeated fluorescence recovery, the proteinosomes were subjected to cycles of bleaching and imaging.

Proteinosomes were prepared as previously described and loaded into capillary glass channels mounted in a Zeiss LSM 880 inverted single-point scanning confocal microscope equipped with a 32 GaAsP photomultiplier tube (PMT) channel spectral detector and imaged using a 40x objective (C-Apochromat 40× NA 1.2 W objective, Zeiss) at 42ºC.

For bleaching of DY530, the 355 nm and 405 nm lasers were used, FITC bleaching was achieved with a 488 nm laser or the 355 nm laser, ROX bleaching

with a 561 nm laser and an additional laser of 405 nm was used. For FRAP experiments the following excitation wavelength and emission ranges were used $\lambda_{DY530}^{exc} = 514$ nm, $\lambda_{DY530}^{emi} = 544$-695 nm; $\lambda_{FITC}^{exc} = 488$ nm, $\lambda_{FITC}^{emi} = 499$–535 nm or $\lambda_{FITC}^{emi} = 490$–553 nm; $\lambda_{ROX}^{exc} = 561$ nm, $\lambda_{ROX}^{emi} = 570$–695 nm. Imaging time varied depending on the region of interest but was typically between 100 ms.frame$^{-1}$ and 300 ms.frame$^{-1}$ for FRAP with 1000 images acquired, up to 10 min to check for fluorescence recovery.

The fluorescence intensity as a function of time for the bleached area, reference and the background were obtained using FIJI and the recovery of the bleached region was normalised against the background and the reference region. An additional normalisation for droplet-based FRAP was undertaken by dividing the fluorescence by the fluorescence of the whole droplet. The kinetic profiles were fit to Eq. 1 using MATLAB to obtain the time constant, τ, of fluorescence recovery or of transport into the droplet (whole droplet FRAP).

$$I(t) = \begin{cases} 1 & \text{for } t < t_0 \\ 1 - A \cdot \exp\left(-\frac{t - t_{bleach}}{\tau}\right) & \text{for } t \geq t_0 \end{cases} \quad (1)$$

Where $t_0$ is defined as the time point immediately after bleaching. The diffusion coefficient is related to the time constant $\tau$ by the relation:[35,36]

$$D = 0.88 \frac{r^2}{4\tau \cdot \log(2)} \quad (2)$$

Here $r$ is the radius of the bleached spot. The diffusion coefficient was averaged over twenty bleaching events across at least two different samples. Interpretation of the apparent diffusivity from photobleaching experiments may be complicated due to the crowding within the proteinosome and two-dimension vs three-dimension diffusion. Despite this, the apparent diffusion coefficients allow us to make comparisons between our studies and should not be treated as absolute values. For repeated bleaching experiments, recovery time was up to 6.4 secs for the polymerase and up to 8.4 s for the DNA. After each photobleach event the dispersion was imaged with the respective excitation and emission wavelengths as described previously. To observe the influx of substrate over time, the proteinosome was identified by bright-field imaging and then changes in fluorescence intensity within the proteinosome was obtained using FIJI. The fluorescence was normalised by the background and by itself. To obtain FRAP information for the polymerase and the DY530-DNA with and without proteinosomes the data was normalised to the background and reference region only.

**Determination of the degree of polymerase sequestration**. The degree of sequestration of FITC labelled polymerase was estimated by determination of the fluorescence intensity of FITC inside and outside of the droplet without background removal.

$$K = \frac{F_{in}}{F_{out}} \quad (3)$$

Where $F_{in}$ is the fluorescence intensity inside the proteinosomes and $F_{out}$ is the fluorescence intensity outside of the proteinosome. To do this, a dispersion of proteinosomes that had been incubated with 10.2 μM (final concentration) of FITC labelled polymerase in a glass capillary channel was imaged using a Zeiss LSM 880 inverted single point scanning confocal microscope equipped with a 32 GaAsP PMT channel spectral detector and a 32-channel Airy Scan detector and imaged using a 40× objective (C-Apochromat 40x NA 1.2 W objective, Zeiss) at 42 °C. A tile scan of 15 images (354.25 × 354.25 μm) was obtained with $\lambda_{FITC}^{exc} = 488$ nm, $\lambda_{FITC}^{emi} = 499$–535 nm. A ROI was found within the proteinosome and outside of the proteinosome and the partition coefficient was determined. It was assumed that there were no effects from changes to the quantum efficiency of FITC within the proteinosome compared to the outside of the proteinosome. The partition coefficient was determined for at least 14 different proteinosomes and the average and standard deviation obtained.

**Confirmation of localisation of PEN DNA reaction within the proteinosomes**. 7.5 μL of supernatant from a solution of sedimented proteinosomes was mixed with 7.5 μL of reaction buffer containing the PEN enzymes and either 0.1 mM dNTPs or no dNTPs at all, then the solutions were incubated at 42 °C using a SPARK 20 M well plate spectrophotometer (TECAN AG, Mannedorf, Switzerland). The change in fluorescence intensity of EvaGreen, was obtained at wavelength/bandwidth $\lambda_{EG}^{exc} = 470/20$ nm, $\lambda_{EG}^{emi} = 515/15$ nm. Additionally, the proteinosome solution was well-mixed before taking 7.5 μL and mixed with the buffer solution. The kinetics of the reaction was measured as described previously.

**PEN DNA reaction- autocatalysis in buffer**. The sequence of template DNA and substrate DNA (Supplementary Table 1) for the autocatalytic reaction was used as described in ref. [31] with a melting temperature of 52.2 °C (50% bound) (at 42 °C, approximately 70% of substrate DNA is bound). The reaction mixture comprised of Buffer A (20 mM Tris-HCl, 10 mM (NH$_4$)$_2$ SO$_4$, 10 mM KCl, 2 mM MgSO$_4$), reaction buffer (50 mM NaCl, 6 mM MgSO$_4$, Synperonic F108 (1 g L$^{-1}$), Netropsin (1 μM)), DTT (3 mM), native BSA (0.5 mg mL$^{-1}$), EvaGreen dye (2.2×), dNTPs

(0.1 mM), primer DNA (1 nM), Nb.BsmI nickase (400 U mL$^{-1}$), Bst DNA polymerase (12.8 U mL$^{-1}$), DNA template and ttRecJ exonuclease. The DNA template and the ttRecJ exonuclease were varied from 0-1000 nM and from 0.6 nM- 9.6 nM respectively.

Autocatalytic PEN DNA reactions were characterised using a SPARK 20 M well plate spectrophotometer (TECAN AG, Mannedorf, Switzerland) by loading 14 µL of the reaction mixture (prepared on ice) into a 384 well plate (Flat black, Greiner) (at room temperature) and then placing the well plate into the well plate reader where the sample chamber had been pre-heated to 42 °C. The fluorescence intensity of bound EvaGreen dye to DNA was measured using wavelength/bandwidth of $\lambda_{exc}$ = 470/20 nm and $\lambda_{em}$ = 515/15 nm every minute for 400 min. Data were collated in Microsoft Excel.

The same experiments were also undertaken using the optical microscope. In this case, 5 µL of reaction mixture was loaded into a glass capillary channel. The sealed capillary channel was loaded onto a pre-heated sample chamber (Tempcontrol 37-2 digital, TePeCon GmbH (Ziegelstraße, Erbach, Germany) at 42 °C on a Zeiss Axiovert 200 M inverted widefield microscope. The microscope was equipped with a 16-channel CooLED pE-4000 and an Andor Zyla PLUS sCMOS camera in phase-contrast mode or in fluorescence. Images with exposure time of 300 ms were taken every 3 mins for at least 500 mins with EvaGreen ($\lambda_{ex}$ = 475/28 nm and $\lambda_{em}$ = 525 nm/45 nm beamsplitter H 488 LPXR) and with exposure time of 150 ms every 15 mins with DY530 ($\lambda_{ex}$ = 542/27 nm and $\lambda_{em}$ = 593/46 nm Beamsplitter H 560 LPXR).

The kinetic profiles from the well plate reader and the optical microscopy experiments were compared for the same samples. To do this, samples were split into two and run on the microscope or the well plate reader. The data were normalised to 1 at the same time point and were in close agreement (Supplementary Fig. 18) demonstrating that bulk measurements in the plate reader correspond to average measurements in the microscope set-up.

**Compartmentalised autocatalytic reactions within proteinosomes**. Proteinosomes prepared either by microfluidics or via bulk methodologies were loaded into a well plate reader for spectroscopic measurement or into a glass capillary for microscopy experiments. Typically, proteinosomes, of known concentration, containing DNA template were incubated with the reaction mixture Buffer A (20 mM Tris-HCl, 10 mM (NH$_4$)$_2$SO$_4$, 10 mM KCl, 2 mM MgSO$_4$) (see Supplementary Table 2); reaction buffer (50 mM NaCl, 6 mM MgSO$_4$, Synperonic F108 (1 g L$^{-1}$), Netropsin (1 µM)) (see Supplementary Table 3); DTT (3 mM); native BSA (0.5 mg mL$^{-1}$); EvaGreen dye (2.2×); mix solution of dNTP (0.1 mM); primer DNA (1 nM); Nb.BsmI nickase (400 U mL$^{-1}$); Bst DNA polymerase (12.8 U mL$^{-1}$); and ttRecJ exonuclease at 42 °C. 14 µL total volume of reaction was loaded into a 384-well plate sealed with a transparent plastic sheet (EASYSEAL, GreinerBio-one) for spectroscopic measurements or 5 µL was loaded into glass capillaries with inner diameter of 1 mm × 100 µm and length of 5 cm (Vitrocom Hollow Rectangle Capillaries) and sealed with epoxy (5 min epoxy, R&G Faserverbundwerkstoffe) to prevent evaporation for microscopy experiments. The number of proteinosomes was kept consistent between experiments (see above). Time-resolved measurements were undertaken on the well plate reader at wavelength/bandwidths, $\lambda_{exc}$ = 470/20 nm and $\lambda_{em}$ = 515/15 nm or with a 10x objective (Plan-Neofluar 10x NA 0.3 Ph1, Zeiss) mounted onto a Zeiss Axiovert 200 M inverted widefield microscope equipped with a 16-channel CooLED pE-4000 and an Andor Zyla PLUS sCMOS camera. The location of the proteinosomes was obtained by imaging the DY530 tagged to the DNA complex using the filter set wavelength/bandwidths, $\lambda_{ex}$ = 542 /27 nm and $\lambda_{em}$ = 593/46 nm, beamsplitter H 560 LPXR. The autocatalytic reaction was tracked by imaging the fluorescence increase of EvaGreen ($\lambda_{ex}$ = 475/28 nm and $\lambda_{em}$ = 525/45 nm and beamsplitter H 488 LPXR) which binds to double-stranded DNA. The capillary channel was imaged with 300 ms exposure time every 3 min for at least 500 min.

To determine the effect of exonuclease concentration on compartmentalised autocatalytic reactions, varying amounts of exonuclease (0-9.6 nM) were added to the reaction mix containing proteinosomes. Kinetics were obtained by using optical microscopy techniques as described previously. In the same way, the effect of DNA template concentration on the autocatalytic rates was characterised by changing the amount of proteinosomes added to the reaction mixture. To obtain the total volume of reaction mixture and concentration of template DNA in the experiment, the total concentration of template DNA was determined by measuring the total volume of proteinosomes within the capillary channel using a custom-written FIJI macro script (code available in the SI, also see the data availability statement) and obtaining the total concentration based on the calibration that each proteinosome contains 0.12 µM of DNA. These experiments have been replicated where the same general trends were observed. However, due to the difficulty in obtaining replicated experiments with the same density of proteinosomes, only a single experiment with determined DNA concentrations have been shown.

**Kinetic analysis for autocatalytic reaction**. To compare the rate of reaction across all datasets, kinetic profiles were fit to an exponential (logistic) model to early time points to obtain the rate constant, $k$, of the autocatalytic reaction. Where $A$ is the fluorescence intensity, $L$ is the maximum of the curve (or 1 for a

normalised curve), $x_0$ is sigmoidal mid-point.

$$A(x) := \frac{L}{1 + e^{k(x-x_0)}} \quad (4)$$

To obtain the data points for the fitting, the data can be normalised and a Gaussian fit to the first differential between each data point as a function of time using the following equation:

$$f(x) := a1.e^{\left(-\left(\frac{x-b1}{c1}\right)^2\right)} \quad (5)$$

$a1$, $b1$ and $c1$ are obtained from the fitting and $x_{max}$ is obtained by rearranging the Gaussian model:

$$x_{positive}(a1, b1, c1, f) := c1.\sqrt{\log\left(\frac{a1}{f}\right)} + b1 \quad (6)$$

The logistic model is then fit from $t = 0$ to the time when $f = (a1)/2$ to obtain the rate constant.

This analysis is based on exponential models which have been used to effectively fit the initial autocatalytic growth and predation of PEN DNA reactions[24]. Here we used a logistic model to fit to a greater number of data points to increase the reliability of the rate constant. When using the logistic model, the range of fitting is limited to the time points where the autocatalytic growth is still described by an exponential growth. This occurs from time zero to when the growth rate reaches a maximum and this rate starts decreasing. In cases where it was not possible to determine the mid-point to decay using this method, the data were fit to a defined time point $t = f$ which provided statistically sufficient data points for analysis.

**Communication network**. To build a two-node network, a second template called trigger (T$_1$) labelled at 5′ with dye ROX and functionalised with Biotin-Teg at 3′ was designed to programme for a linear reaction which would produce a primer strand that was also the substrate for the autocatalytic reaction. Two populations of proteinosomes were prepared using the microfluidic device as described previously where each population of proteinosome contained either T$_1$ or T$_2$ at a concentration of 1 µM. After the proteinosomes were transferred into water via dialysis, the concentration of the proteinosomes was determined and each of the populations were concentrated and mixed together at different ratios into a glass capillary channel. The concentration of population 1 containing T$_1$ was varied whilst the concentration of population 2 containing T$_2$ was kept constant. Populations of proteinosomes were mixed at T$_1$:T$_2$ ratios of 1:1 (0.067 nM:0.068 nM), 0.5:1 (0.035 nM:0.068 nM), and 0.25:1 (0.017 nM:0.068 nM).

Proteinosomes were loaded into capillary channels and the reaction was triggered by the addition of a reaction buffer containing an initial concentration of 50 nM of S$_1$, 12.8 U mL$^{-1}$ polymerase, 400 U mL$^{-1}$ nickase, 0.6 nM exonuclease, 0.1 mM dNTPs and 2.2× EvaGreen and imaged by a 10× objective (Plan-Neofluar 10× NA 0.3 Ph1) mounted onto a Zeiss Axiovert 200M inverted widefield microscope equipped with a 16-channel CooLED pE-4000 and an Andor Zyla PLUS sCMOS camera. The location of the two different populations of proteinosomes were identified by two different DNA- labelling dyes (ROX and DY530: ROX- $\lambda_{ex}$ = 575/15 nm and $\lambda_{em}$ = 641/75 nm Beamsplitter HC BS 596F and DY530- $\lambda_{ex}$ = 542/27 nm and $\lambda_{em}$ = 593/46 nm Beamsplitter H 560 LPXR). The reaction was measured by detection of EvaGreen (EvaGreen - $\lambda_{ex}$ = 475/28 nm and $\lambda_{em}$ = 525/45 nm Beamsplitter H 488 LPXR) as a function of time (every 3 min for approximately 450 min).

Experiments were prepared in three different capillary channels, one experiment (as previously described) and two controls, one with no proteinosomes containing template 1 and one with no dNTPs. Three experiments per condition were imaged with a single frame (or ROI) every time, e.g. three conditions meant 9 frames or 9 experiments. Typically, 13–28 proteinosomes were observed and analysed within the field of view. Droplet segmentation and time-lapse data extraction were executed in the Fiji macro script to obtain the integrated fluorescence intensities divided by the area.

**Reporting summary**. Further information on research design is available in the Nature Research Reporting Summary linked to this article.

# Data availability

The datasets generated during and/or analysed during the current study are available via the following https://doi.org/10.17617/3.6FGVUS. Source data are provided with this paper.

# Code availability

The Fiji script used for microscop data analysis is provided in the Supplementary Methods. The code generated during the current study is available at https://doi.org/10.17617/3.6FGVUS.

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

## Acknowledgements

T.Y.D.T., A.Z. and G.F. acknowledge financial support from the MPG and MaxSynBio Consortium, jointly funded by the Federal Ministry of Education and Research (Germany) and the Max Planck Society and the MPI-CBG (M.G., A.Z.). G.F. was supported by the Deutsche Forschungsgemeinschaft (DFG, German Research Foundation) under Germany's Excellence Strategy – EXC-2068 – 390729961– Cluster of Excellence Physics of Life of TU Dresden (G.F.) and EXC-1056—Center for Advancing Electronics Dresden (G.F.). We acknowledge the Light Microscopy Facility and the Image analysis facility at the MPI-CBG for assistance and Benjamin Friedrich for useful discussions. AdMello acknowledges partial support from a National Research Foundation (NRF) grant funded by the Ministry of Science, ICT and Future Planning of Korea, through the Global Research Laboratory Program (Grant number 2009-00426) (M.U.). We thank the Microstructure Facility of the BIOTEC at the Technische Universitat Dresden (partly funded by the state of Saxony and the European fund for Regional Development – EFRE (100344812). We thank the Center for Macromolecular Structure Analysis in the Leibniz Institute of Polymer Research, Dresden for GPC analysis of the PNIPAAm chain.

## Author contributions

T-Y.D.T. conceived the research. A.Z., G.F., M.F.G., M.U., D.W. contributed to the design of and the undertaking of the experiments. A.Z., G.F., M.F.G., D.W., T.-Y.D.T., analysed the data. All authors contributed to the writing of the manuscript.

## Funding

## Competing interests

The authors declare no competing interests.
