## [Peer Review File · Nature Communications]

REVIEWER COMMENTS

Reviewer #1 (Remarks to the Author):

This work describes the incorporation of PEN (polymerase-exonuclease-nickase) toolbox reactions into proteinosome microcompartments. The authors used a microfluidic droplet maker to form monodisperse proteinosomes and streptavidin-bound PEN template strands were encapsulated in the proteinosomes during the microfluidic assembly process. FRAP measurements show that the template strands were immobile in the interior of the proteinosomes. Autocatalytic PEN reaction where the substrate strands and enzymes diffuse through the compartment membrane was demonstrated. The authors also show a two-population reaction cascade where population 1 implements a linear PEN reaction that produces a signaling strand that activates an autocatalytic PEN reaction in population 2. The work is well described and the use of enzymatic components would allow for a much wider range of spatiotemporal protocell behaviours compared to non-enzymatic variants. As such I recommend publication after the following issues are addressed.

Comments

1) Confocal micrograph on Fig. 1b show that both the protein-polymer conjugate and the fluorescently labeled DNA are non-uniformly distributed in the interior of the proteinosome. What could be the cause of this? In previous publications on proteinosomes the fluorescent protein-polymer conjugate is typically uniformly distributed in the interior.

2) It is assumed that the proteinosomes contain 1 μM of DNA template, which is the concentration that was used during the assembly process. Could you attempt to quantify the DNA template concentration in proteinosomes after they are transferred to water? This would provide valuable information to assess the efficiency of the presented encapsulation method. The previous published method has an encapsulation efficiency of 33%. Additionally, the calculations of reaction rate per DNA template of Fig. 3 seem to be based on this value, which is another reason to verify the localized DNA concentration in proteinosomes.

3) What is also a bit confusing regarding the localization method is that in the main text it is written that the DNA template concentration was 1 μM , while in the materials section it is written "Typically, the aqueous solution consisted of protein-PNIPAAm (4 mg.mL⁻¹), DNA-biotin-streptavidin conjugates (0.5 and 1 μM) was dissolved in 100 mM HEPES buffer at pH 7.2." Also, was there an experimental reason for using 2X excess of streptavidin?

4) On Fig. 3 the reaction rates are given in terms of change in fluorescence intensity rather than change in concentration. For clarity it might be better to then use RFU/min as a unit rather than 1/min.

5) On Fig. 4d the fluorescence trace corresponding to population 1 shows an initial increase followed by decrease. What causes it to decrease? Could it be the recruitment of DNA polymerase to population 2? However the decrease begins around the time that the population 2 trace has plateaued, so this is probably not the reason.

6) The manuscript generally reads well, however it seems to contain rather many typing errors that should be corrected. Just to name a few: The reference to Supplementary Fig. 15 on page 14 doesn't seem to be correct. Same for the reference to Supplementary Fig. 17 on page 22 because Sup. Fig. 17 does not exist in the supplement (should be 15 probably?). And the same for reference to Sup. Fig. 18 on page 25. In the caption of Fig. 3a(iii) the concentration of template was probably 1 μ M and not 1 mM. In Fig. 1b(i) it is written "protein conjugate labelled with carboxy-X-rhodamine (ROX) and DNA complex labelled with fluorescein isothiocyanate (FITC)", I think it was the other way around.

7) I do not fully agree with the following statement in the conclusions: "Previous studies have shown the ability to programme networks based on DNA strand displacement within proteinosomes, however, these systems lack the ability to drive the system out of

equilibrium". This is not true. In general, using fuel-driven DSD reactions it is well possible to drive CRNs into an out-of-equilibrium regime (see for example the DSD oscillator by Winfree et al. Even in proteinosomes this is possible. Is it easier with enzymes? Certainly! I think this claim needs to be rephrased a bit based on my comments above.

I want to congratulate the authors with their work.

With best regards,

Tom de Greef

Reviewer #2 (Remarks to the Author):

This work designed and performed a PEN DNA reaction inside proteinosomes, and characterized the effect of compartmentalization on this biological reaction. The whole study performed very carefully and convincingly. The conclusion that the rates of reaction were increased by an order of magnitude and reaction kinetics were more readily tuneable by enzyme concentrations in proteinosomes compared to buffer solution, was a valuable point for the future building synthetic compartmentalized reaction networks. In general, this is a very interesting study. While, my two main concerns are:

(1) besides the two-node networks design based on programmed proteinosomes, the further demonstration or certain application of such studied biological reaction seems not strong. If possible, the further design to show the significance of this studied biological reaction should be performed.

(2) the basic physical properties of the generated proteinosome are not shown, such as the permeability, especially given it would affect the biological reaction obviously. Does the loading procedure affect the activity of the DNA template? How about the loading efficiency etc.

In addition, there are some typos:

1. As mentioned in page 22, a calibration curve relating the number of proteinosomes to their total intensity of fluorescence was explained in supplementary Figure 17. However, supplementary Figure 17 could not be found in the supporting information part.
2. Full stop was gone at the 14 line in page 10.
3. In supplementary Figure 11, there were several punctuations and word lost such as full stop and that "Polymerase" should be followed after "12.8 units.mL⁻¹"

We thank the editor and the referees for their constructive comments with respect to our manuscript titled „Programmable synthetic cell networks regulated by tuneable reaction rates“. Please find enclosed our response to the comments in a point-by-point fashion with additional experiments and analysis. Every major change and addition to the main manuscript or to the SI has been highlighted in yellow in the resubmission and is also shown here in yellow. Changes that are not directly from the reviewers comments are shown in yellow highlights in the main text and SI. These are all minor typo corrections and are summarised below:

Changes to the abstract:

From “We exploited these properties to regulate the reaction kinetics in two node compartmentalised reaction networks comprised of linear and autocatalytic reactions were built using bottom up approaches.” **To** “We exploited these properties to regulate the reaction kinetics in two node compartmentalised reaction networks comprised of linear and autocatalytic reactions which were established by bottom up synthetic biology approaches.”

Changes to the main text:

From “We also showed reaction rates, as a function of template concentration, were an order of magnitude greater than those observed in buffer solution.” **To** “We also showed **that** reaction rates, as a function of template concentration, were an order of magnitude greater than those observed in buffer solution.”

In Figure 1: From “Dual fluorescence confocal images of proteinosomes” to “dual **colour** fluorescence confocal images of proteinosomes” **and From** “protein conjugate labelled (FITC). To protein conjugate labelled **with FITC**.”

From “Our results show that microfluidics techniques are capable of producing proteinosomes with consistent size and template concentration, offering high levels of control by reducing variability within the population.” **To** “Our results show that microfluidics techniques are capable of producing proteinosomes with consistent size and template concentration, offering high levels of control **and** reducing variability within the population.”

From “As the molecular weight of exonuclease (73 kDa) and nickase (**80 kDa**) are very similar or less than the polymerase” **to** “As the molecular weight of exonuclease (73 kDa) and nickase (**78 kDa**) are very similar or less than the polymerase”

Methods: From “ and obtaining the total concentration based on the assumption that each proteinosome contains 1 μm of DNA.” **To** “ and obtaining the total concentration based on the assumption that each proteinosome contains **0.12 μm** of DNA.”

Additional methods added to the main text:

The synthesis of bis(propylsulfanylthiocarbonyl) disulfide and PNIPAAm were performed according to literature methods.^{30,34} The PNIPAAm was characterized by gel permeation chromatography. The dispersity ($\mathcal{D} = M_w/M_n$; M_n is the number average molecular weight; M_w is the weight average molecular weight of block copolymers) were detected using a Size exclusion column equipped with a MALLS detector (MiniDAWN-LS detector, Wyatt Technology, California, USA) and a viscosity/refractive index (RI) detector (ETA-2020, WGE Dr. Bures, Brandenburg, Germany). The column (PL MIXED-C with a pore size of 5 μm , 300 x 7.5 mm) and the pump (HPLC pump, Agilent 1200 series) were from Agilent Technologies (California, USA). 2% vol water in dimethylacetamide (DMAc) and 3 g/L of LiCl were flowed at a rate of 0.5 mL/min to elute the polymer. Poly(2-vinylpyridin) was used as a standard at 2 mg/mL after filtration through a 0.2 μm filter. The data were processed using Cirrus GPC offline GPC/SEC software (version 2.0). The PNIPAAm chain had an (M_n) of 23000 g/mol, a molecular weight (M_w) of 27000 g/mol resulting in a dispersity index ($\mathcal{D} = M_w/M_n$) of 1.17 unless otherwise stated.

Additional materials added to the SI:

Synperonic[®] F 108 surfactant, DL-Dithiothreitol (DTT, MW. = 154.2 g.mol⁻¹, $\geq 98\%$ TLC and $\geq 99\%$ titration), Netropsin dihydrochloride from *Streptomyces netropsis* powder (MW = 503.4 g.mol⁻¹, $\geq 98\%$, HPLC and TLC), Glucose oxidase from *Aspergillus niger* (GOx, MW = 160 kDa), Bovine Serum Albumin lyophilized powder (BSA, MW = 66 kDa $\geq 96\%$), alcohol dehydrogenase from *Saccharomyces cerevisiae* (ADH, MW = 141 kDa), formate dehydrogenase from *Candida boidinii* (FDH, MW (monomer) = 41 kDa, and 2-ethyl-1-hexanol ($\geq 98\%$), anhydrous dimethyl sulfoxide (MW = 78.1 g.mol⁻¹, $\geq 99.9\%$), N-isopropylacrylamide (NIPAAm, MW = 113.2 g.mol⁻¹, 98 %) and Hexane (MW = 86.2 g.mol⁻¹, reagent grade, $\geq 99\%$), monopotassium phosphate (KH₂PO₄) (MW = 136.1 g.mol⁻¹), Tris-HCl (MW = 121.1 g.mol⁻¹), sodium hydrogen carbonate (MW = 84 g.mol⁻¹), sodium carbonate anhydrous (MW = 106 g.mol⁻¹), chlorotrimethylsilane (MW = 108.6 g.mol⁻¹) and Hellmanex[®] III were purchased from Sigma-Aldrich (Missouri, USA) and used without further purification. Ammonium sulphate ((NH₄)₂SO₄, MW = 132 g.mol⁻¹), potassium chloride (KCl, MW = 74.6 g.mol⁻¹), magnesium sulphate (MgSO₄, MW = 120.4 g.mol⁻¹), sodium chloride (NaCl, MW = 58.4 g.mol⁻¹), Trizma hydrochloride (Tris-HCl, MW = 157.6 g.mol⁻¹), hydrochloric acid (34.46 g.mol⁻¹) and Amicon ultrafilters (10 kDa and 30 kDa cut-off), unconjugated streptavidin from *Streptomyces avidinii* (MW ~ 60 kDa) were all purchased from Merck (Darmstadt, Germany). Amine-reactive fluorescein 5-isothiocyanates isomer I (FITC, MW = 389 Da) and Zeba[™] spin columns (MW cut-off 7 kDa and 40 kDa) and Pierce BCA protein assay kit were purchased from ThermoFisher Scientific (Massachusetts, USA). Methanol and N,N-Dimethylacetamide was purchased from VWR chemicals (Pennsylvania, USA). Toluene (MW = 92.14 g.mol⁻¹) was purchased from Carl Roth GmbH (Karlsruhe, Germany). 3-[Methoxy(polyethyleneoxy) propyl]trimethoxysilane (MW = 1120-1250 g.mol⁻¹) was purchased from abcr GmbH (Karlsruhe, Germany). LiCl was purchased from Honeywell (North Carolina, USA), Poly(2-vinylpyridin) was purchased from Polymer Standards Service, (Mainz, Germany). EvaGreen nucleic acid dye (in water) was obtained from Biotium (Freemont, USA). DNA strands (unmodified, with

phosphorothioate modifications or tagged with biotin-TEG or DY530) were purchased from Biomers GmbH (Ulm, Germany) with HPLC purification. DNA and RNA strands labelled with FAM on the 5' end (CATTCTGACGAG, TCGAGTCTGTT, GCACUUCGGUGC) were purchased from Eurofins Scientific (Luxemburg). Bst.DNA polymerase large fragment (67 kDa, 8000 units/ml, 7.76 µg/ml), Nb.BsmI nickase (~78 kDa, 20.9 µg/ml, 10,000 units.mL⁻¹) and deoxynucleotide solution mix (dNTP, 10 mM each) were purchased from New England Biolabs (NEB, Massachusetts, USA). The polymerase was diluted 5X in storage buffer (Supplementary Table 5) and stored at -20 °C until further use. The recombinant *Thermus thermophilus* ttRecJ exonuclease (73 kDa) was a kind gift from Dr. André Estevez-Torres (CNRS and Sorbonne Université, Paris, France) and produced as described.¹ Monomeric Green fluorescent protein (monoGFP, MW = 25 kDa), were a kind gift from the Protein Expression Purification and characterisation facility (PEPC), MPI-CBG, Dresden, Germany. PEGylated bis(sulfosuccinimidyl)suberate (BS(PEG)₉, MW = 708 g.mol⁻¹) purchased from ThermoFisher Scientific was dissolved in anhydrous dimethyl sulfoxide, aliquoted and stored under argon gas at -20°C until use. EPOXY resin and hardener (1:1 ratio), was purchased from R&G Faserverbundwerkstoffe (Waldenbuch, Germany) and used as directed by the manufacturer's instructions. F-Boden Black 96 well and Flat Black 384 well plates were purchased from Greiner Bio-One, (Kremsmünster, Austria). Hollow rectangle capillaries were purchased from Vitrocom (New Jersey, USA). Silicon wafer was purchased by Silicon Materials, Germany. SU-8 2010 photoresist was purchased from Microchem (Texas, USA) and polydimethylsiloxane was purchased from Dow Corning, (Michigan, USA). Capillary, parafilm based custom made glass slides were prepared as described below with glass slides (ThermoFisher Scientific, Massachusetts, USA, 26 x 76 mm), precision coverglass (Paul Marienfeld GmbH Co. KG, Lauda-Königshofen, Germany, 22 x 22 mm, No. 1.5H), parafilm (Bemis, Wisconsin, USA), and Twinsil quick (Dental-Produktions- und Vertriebs-GmbH, Wipperfürth, Germany). Borosilicate glass hollow rectangular capillaries (length 1 mm, inner diameter 0.1mm) were purchased from CM scientific, Republic of Ireland.

Reviewer #1 (Remarks to the Author):

This work describes the incorporation of PEN (polymerase-exonuclease-nickase) toolbox reactions into proteinosome microcompartments. The authors used a microfluidic droplet maker to form monodisperse proteinosomes and streptavidin-bound PEN template strands were encapsulated in the proteinosomes during the microfluidic assembly process. FRAP measurements show that the template strands were immobile in the interior of the proteinosomes. Autocatalytic PEN reaction where the substrate strands and enzymes diffuse through the compartment membrane was demonstrated. The authors also show a two-population reaction cascade where population 1 implements a linear PEN reaction that produces a signaling strand that activates an autocatalytic PEN reaction in population 2. The work is well described and the use of enzymatic components would allow for a much wider range of spatiotemporal protocell behaviours compared to non-enzymatic variants. As such I recommend publication after the following issues are addressed.

Comments

1) Confocal micrograph on Fig. 1b show that both the protein-polymer conjugate and the fluorescently labeled DNA are non-uniformly distributed in the interior of the proteinosome. What could be the cause of this? In previous publications on proteinosomes the fluorescent protein-polymer conjugate is typically uniformly distributed in the interior.

The reviewer makes a pertinent observation regarding the inhomogeneity of the fluorescently labelled DNA within the interior of the proteinosomes. We typically observe this when protein polymer conjugate has been labelled with FITC. Therefore, it is likely that FITC changes the solubility of the protein-polymer conjugate which causes some aggregation within the proteinosome. In order to reduce this effect and additionally to provide a comparable model system to determine the structural features of the proteinosome the fluorescently labelled protein conjugate was mixed at a ratio of 1:2 with the non-fluorescently labelled protein conjugate.

2) It is assumed that the proteinosomes contain $1 \mu\text{M}$ of DNA template, which is the concentration that was used during the assembly process. Could you attempt to quantify the DNA template concentration in proteinosomes after they are transferred to water? This would provide valuable information to assess the efficiency of the presented encapsulation method. The previous published method has an encapsulation efficiency of 33%. Additionally, the calculations of reaction rate per DNA template of Fig. 3 seem to be based on this value, which is another reason to verify the localized DNA concentration in proteinosomes.

We thank the reviewer for this comment. As a consequence of this comment, we have now quantified the amount of DNA within the proteinosomes after they have been transferred to water. To do this, we undertook a serial dilution of DY530-labelled DNA-biotin streptavidin and measured these at the same microscope settings as those used to image the proteinosomes for the kinetic experiments. Using this calibration curve, we quantified the DNA within the proteinosomes and found that the actual encapsulated DNA concentration was $0.12 \mu\text{M} \pm 0.017 \mu\text{M}$. We have now amended Figure 3, supplementary 13 and the text in the main manuscript with the corrected concentration. In addition, we have included the calibration curve and its methods into the SI as a new supplementary figure 7.

Amendments to existing text and figure captions:

Figure 2: PEN DNA reactions are active within the proteinosomes (a) autocatalytic reaction, when $s = s'$ the PEN DNA reaction is autocatalytic **(b)** (i) Widefield optical fluorescence microscopy images showing increasing EvaGreen fluorescence from DNA production within proteinosomes. Final total concentration of DNA template is 0.06 nM , initial concentration of primer/substrate is 1 nM at $42 \text{ }^\circ\text{C}$. Scale bar is $20 \mu\text{m}$ **(c)** Autocatalytic growth curves obtained from image analysis show fluorescence increase of individual proteinosomes.

Figure 3: Characterisation of autocatalytic PEN DNA reactions within proteinosomes. (a) (i) with an increasing number of proteinosomes and therefore increasing overall template concentration and fixed exonuclease concentration. (a) (ii) Fluorescence widefield microscopy images, at $t = 0$ min and 160 min, showing proteinosome populations at different densities resulting in 0.06 – 0.52 nM total DNA template. Scale bar is 100 μm . (iii) Autocatalytic kinetic curves show the production of DNA inside proteinosomes, 0.06 – 0.52 nM total DNA template at the population densities in (ii). Proteinosomes containing 0.12 μM template were incubated at 42 $^{\circ}\text{C}$ in the reaction buffer containing 0.6 nM exonuclease and triggered with 1 nM of the substrate strand. (b) Reaction rates for autocatalytic reactions carried out with varying template concentration in (i) proteinosomes (blue squares) and (ii) buffer (orange circles). The reactions were triggered with 1 nM primer and incubated at 42 $^{\circ}\text{C}$ in a well-plate reader or imaged in a microscope to obtain the reaction kinetics. The initial rates were obtained from fitting and plot as a function of template concentration. (c) (i) A linear fit of reaction rates for autocatalysis within proteinosomes at varying concentrations of total DNA template gives $392 \pm 104 \text{ min}^{-1} \cdot \text{mM}^{-1}$. Inset shows the residuals from the linear fit. (ii) Bar chart shows the linear fit of autocatalytic reaction rate per template ($\text{d}r/\text{d}T$) for (i) proteinosomes ($392 \pm 104 \text{ min}^{-1} \cdot \text{mM}^{-1}$) (blue), (ii) buffer ($2.3 \pm 1.1 \text{ min}^{-1} \cdot \text{mM}^{-1}$) (red). (d) (i) with constant template concentration and increasing exonuclease concentration, the exonuclease degrades the substrate inside and outside of the proteinosomes. With high exonuclease concentration, the reaction will be completely switched off. (ii) Initial rates of autocatalytic reaction with increasing exonuclease concentration in proteinosomes (blue squares) and a buffer solution (red circles). Data was obtained from at least three repeats for at least three different experiments. The standard deviations (b-d) were obtained from the fit to the data.

Figure 4: Two-node networks based on encapsulated PEN DNA reactions within proteinosomes. (a) The product of a linear reaction programmed into population 1 activates autocatalysis in population 2. (b) Two different populations of proteinosomes are produced by encapsulating two different DNA templates; population 1 contains T_1 labelled with ROX and population 2 contains T_2 labelled with DY530. (c) (i) Dual-channel widefield microscopy show two distinct populations of proteinosomes. Population 1 (blue) containing DNA template T_1 labelled with ROX and population 2 (red) containing DNA template T_2 labelled with DY530. (ii) Time-lapse widefield fluorescence microscopy images show the increase in fluorescence intensity from non-sequence-specific intercalation of Evagreen Dye ($\lambda_{ex} = 475 \pm 28$ nm and $\lambda_{em} = 525 \pm 45$ nm) into the dynamically formed double-stranded DNA due to DNA production inside the proteinosomes for population 1 (blue arrow) and population 2 (red arrow), Scale bar is 20 μ m. (d) Kinetic fluorescence profiles from individual proteinosomes from population 1 (blue curve) and population 2 (red) for a population ratio of 0.5:1 yielding template T_1 and T_2 concentrations of 0.035 nM and 0.068 nM, respectively. (e) Control experiment for the two-node network population. The experiment contained population 2 proteinosomes (final concentration of T_2 was 0.068 nM) and 50 nM of the S_1 primer and no proteinosomes containing T_1 . The plot shows the fluorescence intensity of Evagreen as a function of time obtained from optical microscopy experiments. The red curve is from the proteinosomes containing T_2 and the grey curve is the fluorescence intensity from the background obtained from the area outside of the droplet. (f) Reaction kinetics of a two-node network cell population. Box plot showing the initial rates of the autocatalytic reaction (population 2) as a function of template concentration. Data points in the boxplot come from individual proteinosomes from different experiments 1:1 yielding template T_1 and T_2 concentrations of 0.067 nM and 0.068 nM, respectively, 0.5:1 yielding template T_1 and T_2 concentrations of 0.035 nM and 0.068 nM, respectively and 0.25:1 yielding template T_1 and T_2 concentrations of 0.017 nM and 0.068 nM, respectively. The shaded colours in the plots in a-c are the standard deviations from the kinetic curves from individual proteinosomes in their corresponding populations. Data was from three different experiments from at least 11 proteinosomes.

The modifications to the main text are shown below:

From

“To this end, proteinosomes containing DNA template (1 μM) coding for an autocatalytic reaction,” **To** “The DNA content within the proteinosome was determined by calibration (see supplementary materials and supplementary figure 6) to be 0.12 μM +/- 0.017 μM after the addition of 1 μM of DNA complex. “

From “As the concentration of template DNA within the proteinosome was fixed at 1 μM , the total concentration of DNA template in the dispersion was varied by altering the density of proteinosomes and was typically between 0.5-5 nM of DNA (Figure 3ai). The total concentration of template DNA in the dispersion was determined by measuring the total volume of proteinosomes within the capillary channel and obtaining the total concentration based on the assumption that each proteinosome contains 1 μM of DNA.”

To : “As the concentration of template DNA within the proteinosome was fixed at 0.12 μM , the total concentration of DNA template in the dispersion was varied by altering the density of proteinosomes and was typically between 0.06 – 0.5 nM of DNA (Figure 3ai). The total concentration of template DNA in the dispersion was determined by measuring the total volume of proteinosomes within the capillary channel and obtaining the total concentration based on the assumption that each proteinosome contains 0.12 μM of DNA (supplementary figure 6 and 10). “

From : “The results show that in a dispersion of proteinosomes ($[\text{DNA template}]_{\text{total}} = 0.5 \text{ nM}$)” to The results show that in a dispersion of proteinosomes ($[\text{DNA template}]_{\text{total}} = 0.06 \text{ nM}$).”

From: “high local concentration of template at 1 μM but a low total concentration (nM) in the total dispersion.” **To**: “high local concentration of template at 0.12 μM but a low total concentration (nM) in the total dispersion.”

From” The concentration of population 2 was kept constant whilst the concentration of population 1 was varied (Figure 4f, Supplementary figure 16) at $T_1:T_2$ ratios of 1:1 (0.56 nM : 0.57 nM) (Supplementary figure 16), 0.5:1 (0.29 nM : 0.57 nM) (Figure 4d), and 0.25:1 (0.14 nM : 0.57 nM)” **To** : The concentration of population 2 was kept constant whilst the concentration of population 1 was varied (Figure 4f, Supplementary figure 17) at $T_1:T_2$ ratios of 1:1 (0.067 nM : 0.068 nM) (Supplementary figure 17), 0.5:1 (0.035 nM : 0.067 nM) (Figure 4d), and 0.25:1 (0.017 nM : 0.068 nM) (Supplementary figure 17).

Methods:

From “To do this, proteinosomes containing 1 μM of DNA template” **To** “(Supplementary Figure 10). To do this, proteinosomes containing 0.12 μM of DNA”

From “obtaining the total concentration based on the calibration that each proteinosome contains 1 μM of DNA” to: “obtaining the total concentration based on the calibration that each proteinosome contains 0.12 μM of DNA”

From “1:1 (0.56 nM : 0.57 nM), 0.5:1 (0.29 nM : 0.57 nM), and 0.25:1 (0.14 nM : 0.57 nM).”

To : “of 1:1 (0.067 nM : 0.068 nM), 0.5:1 (0.035 nM : 0.068 nM), and 0.25:1 (0.017 nM : 0.068 nM).”

Supplementary figure 15. Autocatalytic kinetics in proteinosomes at different exonuclease concentrations. Proteinosomes containing 0.12 μM template were incubated in the presence of 1 nM DNA primer at 42 °C in a Spark 20 M well plate reader spectrophotometer (Tecan AG, Männedorf, Switzerland) using excitation and emission wavelengths of 470 ± 20 nm and 515 ± 15 nm respectively.

New Methods in the SI:

Determination of DNA concentration within proteinosomes after microfluidic encapsulation.

To determine the encapsulation efficiency of DNA-biotin-streptavidin complex within proteinosomes prepared by microfluidics a calibration curve for DY530- labelled DNA- biotin-streptavidin was undertaken using the Zeiss LSM 880 single point scanning confocal microscope using a 40x objective (C-Apochromat 40x NA 1.2W objective, Zeiss). To do this, a serial dilution of DY530-labeled DNA-biotin-streptavidin at concentrations of 0, 0.031, 0.063, 0.13, 0.25, 0.5 μM DNA were prepared in water and loaded into custom made channel slides. The samples were loaded onto the microscope and incubated at 25° C for 30 mins to ensure temperature equilibration. The microscope settings were consistent with those used for the autocatalytic and linear kinetic experiments (Figure 3) ($\lambda_{\text{DY530}}^{\text{exc}} = 514$ nm $\lambda_{\text{DY530}}^{\text{emi}} = 543-695$ nm) and for the proteinosomes prepared for the effect of encapsulation on the autocatalytic reaction respectively.

The images were analysed using a FIJI macro code and the average fluorescence intensity at known DY530-DNA-biotin-streptavidin concentration was obtained and plotted. The data was fit to a linear regression equation to obtain the following equation Fluorescence (a.u.) = 4054.4 x (μM) +13.13, ($R^2=0.9975$) and used to convert fluorescence intensity within proteinosomes into concentrations. A Gaussian fit was applied to the histogram of number of proteinosomes vs concentration to obtain the average concentration and a standard deviation from 94 proteinosomes. (0.12 μM with a std=0.017 μM).

Supplementary Figure 6: Calibration of DNA concentration for proteinosomes produced by microfluidics. (Left) Serial dilution of DY530-labelled DNA-biotin streptavidin under the same microscope settings as the autocatalytic and linear kinetic experiments with proteinosomes. **(right)** Histogram showing the proteinosomes produced for reactions shown in Figure 3 and 4. Fluorescence units were converted to concentration using the calibration curve and its linear regression and a gaussian fit to the histogram of number of proteinosomes was undertaken to obtain the average concentration 0.12 μM +/-0.02 μM .

3)What is also a bit confusing regarding the localization method is that in the main text it is written that the DNA template concentration was 1 μM , while in the materials section it is written “Typically, the aqueous solution consisted of protein-PNIPAAm (4 mg.mL⁻¹), DNA-biotin-streptavidin conjugates (0.5 and 1 μM) was dissolved in 100 mM HEPES buffer at pH 7.2.” Also, was there an experimental reason for using 2X excess of streptavidin?

We thank the reviewer for this comment. For all of the experiments, 1 μM of total template was added to the dispersion. We aimed to have on average 2 DNA templates per streptavidin which has 4 binding sites to ensure that all of the DNA was bound to the streptavidin. We have modified to text to remove the errors.

” DNA-biotin-streptavidin conjugates (0.5 and 1 μM) was dissolved in 100 mM HEPES buffer at pH 7.2” to
DNA-biotin-streptavidin conjugates (1 μM -DNA) was dissolved in 50 mM HEPES buffer at pH 7.6.

And in the materials and methods from:

“The final concentration of ssDNA was 10 μM and for streptavidin it was 5 μM to achieve a final molar ratio of 2:1 Streptavidin:DNA.”

To : “The final concentration of ssDNA was 10 μM and for streptavidin it was 5 μM to achieve a final molar ratio of 1:2 Streptavidin:DNA.”

And in the main text from : “final molar ratio of 2:1 streptavidin:DNA” **To:** “final molar ratio of 1:2 streptavidin:DNA”

4) On Fig. 3 the reaction rates are given in terms of change in fluorescence intensity rather than change in concentration. For clarity it might be better to then use RFU/min as a unit rather than 1/min.

We thank the reviewer for this comment and have replotted, figure 3 and 4 and supplementary figure 9 (now supplementary 12) as requested.

Figure 3: Characterisation of autocatalytic PEN DNA reactions within proteinosomes. (a) (i) with an increasing number of proteinosomes and therefore increasing overall template concentration and fixed exonuclease concentration. (a) (ii) Fluorescence widefield microscopy images, at $t = 0$ min and 160 min, showing proteinosome populations at different densities resulting in 0.06 – 0.52 nM total DNA template. Scale bar is 100 μm . (iii) Autocatalytic kinetic curves show the production of DNA inside proteinosomes, 0.06 – 0.52 nM total DNA template at the population densities in (ii). Proteinosomes containing 0.12 μM template were incubated at 42 $^{\circ}\text{C}$ in the reaction buffer containing 0.6 nM exonuclease and triggered with 1 nM of the substrate strand. (b) Reaction rates for autocatalytic reactions carried out with varying template concentration in (i) proteinosomes (blue squares) and (ii) buffer (orange circles). The reactions were triggered with 1 nM primer and incubated at 42 $^{\circ}\text{C}$ in a well-plate reader or imaged in a microscope to obtain the reaction kinetics. The initial rates were obtained from fitting and plot as a function of template concentration. (c) (i) A linear fit of reaction rates for autocatalysis within proteinosomes at varying concentrations of total DNA template gives $392 \pm 104 \text{ min}^{-1} \cdot \text{mM}^{-1}$. Inset shows the residuals from the linear fit. (ii) Bar chart shows the linear fit of autocatalytic reaction rate per template (dr/dT) for (i) proteinosomes ($392 \pm 104 \text{ min}^{-1} \cdot \text{mM}^{-1}$) (blue), (ii) buffer ($2.3 \pm 1.1 \text{ min}^{-1} \cdot \text{mM}^{-1}$) (red). (d) (i) with constant

template concentration and increasing exonuclease concentration, the exonuclease degrades the substrate inside and outside of the proteinosomes. With high exonuclease concentration, the reaction will be completely switched off. **(ii)** Initial rates of autocatalytic reaction with increasing exonuclease concentration in proteinosomes (blue squares) and a buffer solution (red circles). Data was obtained from at least three repeats for at least three different experiments. The standard deviations (b-d) were obtained from the fit to the data.

Figure 4: Two-node networks based on encapsulated PEN DNA reactions within proteinosomes. (a) The product of a linear reaction programmed into population 1 activates autocatalysis in population 2. **(b)** Two different populations of proteinosomes are produced by encapsulating two different DNA templates; population 1 contains T_1 labelled with ROX and population 2 contains T_2 labelled with DY530. **(c)** (i) Dual-channel widefield microscopy show two distinct populations of proteinosomes. Population 1 (blue) containing DNA template T_1 labelled with ROX and population 2 (red) containing DNA template T_2 labelled with DY530. **(ii)** Time-lapse widefield fluorescence microscopy images show the increase in fluorescence intensity from non-sequence-specific intercalation of Evagreen Dye ($\lambda_{ex} = 475 \pm 28$ nm and $\lambda_{em} = 525 \pm 45$ nm) into the dynamically formed double-stranded DNA due to DNA production inside the proteinosomes for population 1 (blue arrow) and population 2 (red arrow), Scale bar is 20 μm . **(d)** Kinetic fluorescence profiles from individual proteinosomes from population 1 (blue curve) and population 2 (red) for a population ratio of 0.5:1 yielding template T_1 and T_2 concentrations of 0.035 nM and 0.068 nM, respectively. **(e)** Control experiment for the two-node network population. The experiment contained population 2 proteinosomes (final concentration of T_2 was 0.068 nM) and 50 nM of the S_1 primer and no proteinosomes containing T_1 . The plot shows the fluorescence intensity of EvaGreen as a function of time obtained from optical microscopy experiments. The red curve is from the proteinosomes containing T_2 and the grey curve is the fluorescence intensity from the background obtained from the area outside of the droplet. **(f)** Reaction kinetics of a two-node network cell population. Box plot showing the initial rates of the autocatalytic reaction (population 2) as a function of template concentration. Data points in the

boxplot come from individual proteinosomes from different experiments 1:1 yielding template T_1 and T_2 concentrations of 0.067 nM and 0.068 nM, respectively, 0.5:1 yielding template T_1 and T_2 concentrations of 0.035 nM and 0.068 nM, respectively and 0.25:1 yielding template T_1 and T_2 concentrations of 0.017 nM and 0.068 nM, respectively. The shaded colours in the plots in a-c are the standard deviations from the kinetic curves from individual proteinosomes in their corresponding populations. Data was from three different experiments from at least 11 proteinosomes.

Supplementary figure 12. Linear fits of reaction rates as a function of template for reactions (i) buffer (ii) Residuals are plotted below the linear fits.

5) On Fig. 4d the fluorescence trace corresponding to population 1 shows an initial increase followed by decrease. What causes it to decrease? Could it be the recruitment of DNA polymerase to population 2? However the decrease begins around the time that the population 2 trace has plateaued, so this is probably not the reason.

The increase and decrease in fluorescence intensity for population 1 in figure 4d, arises from degradation of the substrate (S_1) and the product (S_2). These DNA primers are not protected by phosphothiorionate groups which means that they can be degraded by exonuclease. As exonuclease degrades substrate (S_1) the fluorescence intensity within the proteinosomes will decrease as there is no double strand formation in population 1. However, production of S_2 is sufficient to initiate the autocatalytic reaction in population 2 by communication. Our results show that the plateau of population 2 (Figure 4 and SI figure 15) is dependent on the starting concentration of template which determines the concentration of S_2 as a function of time.

The following text has been added to the figure legend of supplementary figure 17 which correlates to figure 4 for additional clarity.

„The decrease in the fluorescence intensity from population 1 is due to the action of exonuclease on the primer strand.“

6)The manuscript generally reads well, however it seems to contain rather many typing errors that should be corrected. Just to name a few: The reference to Supplementary Fig. 15 on page 14 doesn't seem to be correct. Same for the reference to Supplementary Fig. 17 on page 22 because Sup. Fig. 17 does not exist in the supplement (should be 15 probably?). And the same for reference to Sup. Fig. 18 on page 25. In the caption of Fig. 3a(iii) the concentration of template was probably 1 μ M and not 1 mM. In Fig. 1b(i) it is written "protein conjugate labelled with carboxy-X-rhodamine (ROX) and DNA complex labelled with fluorescein isothiocyanate (FITC)", I think it was the other way around.

These typos have been corrected and the manuscript has been checked for further typos and amended.

7) I do not fully agree with the following statement in the conclusions: "Previous studies have shown the ability to programme networks based on DNA strand displacement within proteinosomes, however, these systems lack the ability to drive the system out of equilibrium". This is not true. In general, using fuel-driven DSD reactions it is well possible to drive CRNs into an out-of-equilibrium regime (see for example the DSD oscillator by Winfree et al. Even in proteinosomes this is possible. Is it easier with enzymes? Certainly! I think this claim needs to be rephrased a bit based on my comments above.

WE have rephrased the sentence according to the comments from "Previous studies have shown the ability to programme networks based on DNA strand displacement within proteinosomes, however, these systems lack the ability to drive the system out of equilibrium. Whilst the systems we have presented reach a steady state, PEN DNA reactions are capable of driving out-of-equilibrium reactions which is possible by exploiting and tuning the concentration of exonuclease."

To

"Previous studies have shown the ability to programme networks based on DNA strand displacement within proteinosomes, however, the addition of enzymes will greatly facilitate the ability to drive the systems into an out-of-equilibrium state."

I want to congratulate the authors with their work.

With best regards,

Tom de Greef

Reviewer #2 (Remarks to the Author):

This work designed and performed a PEN DNA reaction inside proteinosomes, and characterized the effect of compartmentalization on this biological reaction. The whole study performed very carefully and convincingly. The conclusion that the rates of reaction were increased by an order of magnitude and reaction kinetics were more readily tuneable by enzyme concentrations in proteinosomes compared to

buffer solution, was a valuable point for the future building synthetic compartmentalized reaction networks. In general, this is a very interesting study. While, my two main concerns are:

(1) besides the two-node networks design based on programmed proteinosomes, the further demonstration or certain application of such studied biological reaction seems not strong. If possible, the further design to show the significance of this studied biological reaction should be performed.

We thank the reviewer for this comment. We agree with the reviewer we have not shown anything beyond the two node network for these programmed proteinosomes. However, as the reviewer themselves highlights we show the effect of compartmentalization on the PEN DNA reaction. They also highlight that this could be important for building synthetic compartmentalised reactions in the future. We show the two node network as a demonstration and potential of using this to build networks but the demonstration of this is beyond the scope of this study. In the introduction and conclusion we have clearly stated the focus and implications of the study *i.e.* that compartmentalization can affect the PEN DNA reaction with advantageous outcomes in the text shown below.

In the introduction:

“These features are important for the rational design of compartmentalised reaction networks and therefore provide a framework for furthering our understanding of reaction dynamics in compartments. This is a fundamental property of biological systems and is crucial for the realisation of synthetic micro-compartments for applications in industrial and engineering applications.”

And in the conclusion:

“Whilst it is still a challenge to probe the effect of compartmentalisation in biological cellular systems, the ability to build micron sized compartments encapsulating enzyme reactions has offered an unique opportunity to address this challenge without biological complexity. Our work represents an important step in bottom-up synthetic biology approaches by combining it with quantitative approaches. As a consequence, we have shown that the kinetic landscape of compartmentalised reactions are not equivalent to free solution. Our results show that it is imperative, that biological behaviours based on networks of chemical reactions have to be considered within the context of compartmentalisation. Simplified physical models of compartments are ideal for exploring these fundamental questions.”

Our studies suggest that utilizing simple physical models of compartments can be effective in considering the effect of compartments on biological reactions and can be an energetically cost effective strategy to increase rates of reaction with important implications for the design and synthesis of synthetic cellular systems compartmentalized reaction networks. The further design of more complex systems goes beyond the scope of this work and would be the focus of further studies. However, there are examples in the literature of more complex networks based on PEN DNA which provide a simple analogue to in-vivo networks such as genetic circuits based on transcription and translation and as computational systems and which can provide analogues to biological systems. We have included the following references to provide some background for the use of synthetic biology approaches to use biological reaction networks and circuits for computing and specifically the use of PEN DNA reactions to for complex networks such as oscillatory predator-prey systems where the authors discuss analogies to biological systems.

2. Dalchau, N. *et al.* Computing with biological switches and clocks. *Nat. Comput.* **17**, 761–779 (2018).

28. Montagne, K., Plasson, R., Sakai, Y., Fujii, T. & Rondelez, Y. Programming an in vitro DNA oscillator using a molecular networking strategy. *Mol. Syst. Biol.* **7**, 466 (2011).
29. Fujii, T. & Rondelez, Y. Predator–Prey Molecular Ecosystems. *ACS Nano* **7**, 27–34 (2013).

Following on from the reviewer's comment we acknowledge that it is important to highlight in our work, that these systems can provide a link to biological systems. In particular, our minimal systems are most ideally suited to provide insights into the physical functionality of biological compartmentalisation. Specific examples could include biological compartments which contain reactions but allow passive diffusion of molecules such as in the outer membrane of mitochondria or in the membranes of gram negative bacteria. We have added additional text in the conclusion to describe this and offer some areas where our study could be interesting for biological systems.

“Moreover, these minimal models could be effective as physical models of biological compartmentalization where membrane bound compartments which contain reactions allow the passive diffusion of molecules such as in mitochondria or in gram negative bacteria.”

In addition, as an outlook in the conclusion, we describe the potential for the use of these systems for further applications

“Compartmentalisation also offers the ability to spatially localise, alter reaction kinetics and reduce the total amount of DNA template required for the reaction. Coupled with the coding ability of DNA, this platform allows access to a large combinatorial space. Our toolkit opens many possibilities for spatial multiplexing using a flexible and modular system, thus providing a general route for building synthetic compartmentalised reaction networks, based on reaction diffusion mechanisms and a minimal number of components.”

(2) the basic physical properties of the generated proteinosome are not shown, such as the permeability, especially given it would affect the biological reaction obviously. Does the loading procedure affect the activity of the DNA template? How about the loading efficiency etc.

We thank the reviewer for this comment. In response we have undertaken additional experiments which have determined the permeability of a range of different molecules into the proteinosomes. Together with the data already presented in the manuscript that show FRAP analysis of the proteinosomes these experiments provide the basis physical properties of the proteinosomes. Specifically, these results give information on the structure of the proteinosomes, the rates of diffusion of the encapsulated DNA, protein conjugate and free DNA within the proteinosomes and the permeability of the proteinosome. Our additional results on permeability confirm that the proteinosomes are permeable to proteins (up to 140 kda). For completeness, we undertook the permeability experiments with different DNA templates and different dyes i.e. ROX-T₁-Biotin, AF594-T₂-Biotin as we had different populations of proteinosomes in our manuscript. AF594-T₂ was used in lieu of DY530-T₂ to ensure sufficient separation of fluorescence emission spectra between FITC labelled proteins and the DNA (See rebuttal (below, main text and supplementary

information (methods, supplementary figure 5 and supplementary table 1). Our results show that the partitioning of different molecules was not affected by either the DNA template sequence or the dye. The results tell us that the concentration of molecules which partition from the external solution into the interior of the proteinosome will be comparable for proteinosomes with different encapsulated biochemistry.

Further to this and as suggested by the reviewer, we determined the effect of the loading procedure (bulk methods vs microfluidics) on the activity DNA within the proteinosome. Our results show that proteinosomes prepared by emulsion methodologies and microfluidic methodologies at the same concentrations have the same activity when the DNA concentrations are equivalent. To do this, we prepared proteinosomes containing DY 530 labelled - T₂ autocatalytic template by microfluidics and by bulk methodologies. Using the fluorescence intensity from Dy530 obtained by fluorescence spectroscopy it was possible to achieve equivalent concentration of DNA encapsulated proteinosomes prepared by the two different methodologies and compare their autocatalytic activities. This was done for two different fluorescent outputs or concentrations of DNA. In both cases, our results show that the activity of the DNA is comparable between the proteinosomes prepared by two different methods (See rebuttal Part 2: Figure R2).

Whilst these results are interesting, we have not included these into the manuscript or the supplementary information as the focus of the work is on the microfluidically prepared proteinosomes. The reason why we present the microfluidically prepared proteinosomes, *only*, is because our previous published work and the results that we present in supplementary figure 2 show that proteinosomes prepared by microfluidics are more consistent in size and DNA encapsulation. We have additionally and subsequently confirmed this in repeat experiments where we have calibrated the DNA concentration in proteinosomes prepared by bulk vs microfluidic methodologies (See rebuttal part 2: Figure R3). Our results showed that the encapsulation efficiency of DNA (starting concentration 1 μ M) into proteinosomes prepared by microfluidics is lower with a smaller distribution 278 +/- 19 nM (RSD = 6%) compared to bulk methods 370 +/- 88 nM (20%) as expected. The fact that proteinosomes prepared by microfluidics are more consistent than bulk methods was further evident after measuring the partition coefficient of GFP within proteinosomes prepared by bulk methods. The partition coefficient was measured for over 30 proteinosomes and the partition coefficient was the same within error between bulk ($K = 0.66 \pm 0.44$) and microfluidically ($K = 0.97 \pm 0.1$) prepared proteinosomes with a much higher standard deviation in bulk prepared proteinosomes. Taken together, there appears to be no difference in the PEN activity or partitioning of molecules into bulk vs by microfluidics prepared proteinosomes with the DNA and partitioning of molecules exhibiting a lower RSD in microfluidically prepared proteinosomes across a population making these systems more suitable for characterizing reactions in populations of compartments.

With regard to the loading efficiency of the DNA into proteinosomes. We quantified the DNA concentration within the proteinosomes utilized for the experiments described in the manuscript. Our calibration showed that the encapsulation efficiency was 12% +/- 2%. As a consequence, we have corrected all of the concentrations of DNA in the manuscript accordingly (See section Part 3 below). It is pertinent to note that there can be differences in encapsulation efficiencies between microfluidically

prepared proteinosomes, for example, we obtain a range of encapsulation efficiency from 12% (See Rebuttal part 3: Figure R4 and Supplementary figure 6) to 28 % (Figure R3). The variability in loading efficiency can be attributed to the efficiency of DNA-biotin-streptavidin crosslinking to the proteinosomes which can be dependent on the efficiency of BS(PEG)₉ crosslinker which is known to degrade over time or to batch-to-batch variability of protein to polymer ratio or even to polymer size. We have not included these findings as the work does not focus on comparing the preparation of the proteinosomes but the effect of the proteinosomes on PEN DNA reactions.

Rebuttal Part 1: Determination of partition coefficients into proteinosomes

Additional text to the main text:

Indeed, determination of the partition coefficient (K) of a range of proteins from monomeric green fluorescent protein (mon GFP, MW = 25 kDa, K = 0.99 ± 0.17), Bovine Serum Albumin (BSA, MW = 67 kDa, K = 0.78 ± 0.17), ttRecJ exonuclease (MW=73 kDa, K = 0.87 ± 0.06) and alcohol dehydrogenase (MW = 141 kDa, K = 1.01 ± 0.17) within proteinosomes loaded with AF594-T₂-Biotin gave partition coefficients of greater than 0.71 for all proteins (See supplementary methods, Supplementary figure 5 and supplementary table 7). Our results also show that the DNA sequence or its fluorescence label does not affect the partition coefficient, within error, indicating that the all molecules will sequester to the same degree proteinosomes encapsulating different proteinosomes prepared by microfluidics. Taken together, our results show that template DNA is isolated and fixed within the proteinosomes and that the enzymes and primer DNA can freely diffuse in and out of the proteinosomes. Specifically, the proteinosomes are permeable to proteins up to 141 kDa regardless of the sequence or the fluorophore encapsulated into the proteinosome.

Supplemental information (method and results):

Determination of the degree of molecular sequestration

The degree of sequestration of Fluorescein Isothiocyanate (FITC) -labelled Alcohol dehydrogenase (ADH), Bovine Serum Albumin (BSA), Formate dehydrogenase (FDH), ttRecJ, GFP, FAM-labelled DNA (CATTCTGACGAG, TCGAGTCTGTT) was estimated by determination of the fluorescence intensity of FITC inside and outside of the droplet with background removal from proteinosome only with the DNA/streptavidin constructs.

$$K = \frac{F_{in}}{F_{out}} \quad (3)$$

Where F_{in} is the fluorescence intensity inside the proteinosomes and F_{out} is the fluorescence intensity outside of the proteinosome.

To do this, proteins were labelled with FITC by firstly, exchanging the buffer of the proteins into into freshly prepared sodium hydrogen carbonate buffer (0.2 M, pH 8.4) and then mixed with a 50x molar excess of FITC. The solution was incubated in the dark for 2 hrs with rotation to ensure efficient mixing. The unbound dyes were then removed from the solution by exchanging the buffer with Trizma[®] Hydrochloride (0.1 M, pH 8.0). Buffer exchange steps were performed with Zeba[™] spin columns (7 kDa

cut off for BSA and FDH or 40 kDa cut off for all other proteins). The final concentration of the protein and the protein to dye ratio was determined by UV-vis absorbance using the Nanodrop 1000 (Thermofisher scientific, Massachusetts USA) at 280 nm and 494 nm.

To determine the sequestration of fluorescently labelled molecules into proteinosomes. Proteinosomes containing ROX-labelled T₁ DNA-biotin-streptavidin complexes or Alexa Fluor 594 (AF594)-labelled T₂ DNA-biotin streptavidin complexes were diluted into a reaction solution without EvaGreen or PEN enzymes. AF594 T₂ DNA-biotin streptavidin complexes were used in lieu of DY530 labelled DNA due to the overlap in fluorescence excitation and emission with FITC labelled substrates. Fluorescent molecules were added to the dispersion of proteinosomes to following final total concentrations (ADH (0.4 μM, 0.06 mg.mL⁻¹), BSA (2.58 - 38.64 μM, 0.17 – 2.55 mg.mL⁻¹), FDH (4.55 μM , 0.2 mg.mL⁻¹), ttRecJ (1.23 μM, 0.09 mg.mL⁻¹), and monoGFP (2 - 20 μM, 0.05 - 0.5 mg.mL⁻¹) and FAM labelled DNA (CATTCTGACGAG, TCGAGTCTGTT, 10 μM) to achieve a final reaction solution of 1x. After 5 mins of incubation, the samples were loaded in homemade silanated customised capillary channels prepared on glass slides. Both ends of the channel were sealed with Twinsil quick two-component silicon glue and incubated at room temperature to allow curing of the glue and then at 42°C for 10 minutes on the microscope stage before imaging.

The samples were imaged using a Zeiss LSM 880 inverted single point scanning confocal microscope equipped with a 32 GaAsP PMT channel spectral detector and a 32-channel Airy Scan detector and imaged using a 20x objective (Plan-Apochromat 20x/ 0.8 objective, Zeiss) at 42°C. A tile scan of 3 by 5 or 4 by 4 images (354.25 x 354.25 μm) was obtained with 10% overlap with $\lambda_{\text{ROX/AF594}}^{\text{exc}} = 561 \text{ nm}$ $\lambda_{\text{ROX/AF594}}^{\text{emi}} = 590-650 \text{ nm}$, $\lambda_{\text{FITC / FAM / GFP}}^{\text{exc}} = 488 \text{ nm}$ $\lambda_{\text{FITC / FAM / GFP}}^{\text{emi}} = 490-560 \text{ nm}$.

Optical microscopy images were analysed using Fiji². A Fiji script was used to detect proteinosomes based on the fluorescence from the ROX-labelled DNA (see Fiji code). Extracted ROIs were used to extract signals from both ROX and FITC/FAM/GFP channel. Partitioning coefficient was calculated by fluorescent signal inside proteinosome divided by fluorescent signal in the buffer with background subtraction using proteinosomes containing ROX-labelled T₁ DNA-biotin-streptavidin complexes. The partition coefficient was determined for at least 30 different proteinosomes and the average and standard deviation obtained.

Supplementary figure 5. Partitioning assay of proteinosome with fluorescently labelled molecules. The experiment was performed by incubating ROX-T₁ or AF594-T₂ (dye labelled DNA templates) encapsulated proteinosomes with fluorescently-labelled molecules (alcohol dehydrogenase (ADH), BSA, Formate dehydrogenase (FDH), monoGFP, DNA oligos, ttRECJ) in the reaction buffers at 42°C. Confocal microscopy images showing the FITC/GFP channel for the molecular partitioning for the control experiment, no fluorescently labelled molecules, BSA-FITC (0.3mg/ml (AF594-T₂), 0.51 mg/ml (ROX-T₁), mono-GFP (0.5 mg/ml), FAM-TCGAGTCTGTT (2.5 μM) and ttRECJ-FITC (0.09 mg/ml). Scale bar is 30 μm. The number average molecular weight (M_n) of PNIPAAm used to prepare these proteinosomes was 15000 g/mol, the molecular weight (M_w) 17000 g/mol and its dispersity index (Đ =M_w/M_n) was 1.13. The ratio of protein: PNIPAAm was 1: 5.7.

Sample name	mw (kDa)	ROX-T ₁ -Biotin		AF594-T ₂ -Biotin	
		Concentration	Partitioning Coefficient	Concentration	Partition coefficient
ADH-FITC	141	0.06 mg/ml	1.09 ± 0.10	0.06 mg/ml	1.01 ± 0.17
BSA-FITC	67	2.55 mg/ml	0.97 ± 0.12	0.3 mg/ml	0.78 ± 0.17
BSA-FITC	67	0.51 mg/ml	0.84 ± 0.04	0.06 mg/ml	0.76 ± 0.16
BSA-FITC	67	0.17 mg/ml	0.71 ± 0.05	0.02 mg/ml	0.80 ± 0.10
FDH-FITC	41*	0.2 mg/ml	1.08 ± 0.07	0.2 mg/ml	1.08 ± 0.07
monoGFP	25	0.5 mg/ml	0.97 ± 0.05	0.5 mg/ml	0.99 ± 0.17
monoGFP	25	0.17 mg/ml	0.97 ± 0.10	0.17 mg/ml	0.96 ± 0.07
FAM-CATTCTGACGAG	4	2.5 μM	0.98 ± 0.06	2.5 μM	0.93 ± 0.06
FAM-TCGAGTCTGTT	4	2.5 μM	0.96 ± 0.11	2.5 μM	1.04 ± 0.10
ttRecJ-FITC	73	0.09 mg/ml	0.98 ± 0.06	0.09 mg/ml	0.87 ± 0.06

*monomer unit

Supplementary table 7: Summary of partition coefficients for a range of different proteins of different molecular weights for proteinosomes containing ROX-T₁-Biotin or AF594-T₂-Biotin obtained by confocal fluorescence microscopy.

Rebuttal Part 2: Comparison between DNA activity in proteinosomes prepared by bulk and microfluidic methods

Figure R2: Effect of loading efficiency on activity of the DNA template at two different proteinosome densities (a, b). Proteinosomes containing the T_2 autocatalytic template tagged with DY530 were prepared using either (i) microfluidics or (ii) bulk methodologies. The proteinosomes were loaded into a well plate and imaged using widefield fluorescence microscopy (10 x objective and filter set with $\lambda_{em} = 542/27$ nm and $\lambda_{ex} = 593 / 46$, nm dichro 560LP) to verify the presence of proteinosomes, Scale bar = 50 μ m (i, ii). (iii) The fluorescence intensity of the DY530 was then determined by fluorescence spectroscopy obtained to determine the DNA concentration in the well plate ($\lambda_{exc} = 539$ nm \pm 7.5 nm and $\lambda_{em} = 561$ nm \pm 7.5 nm). (iv) The autocatalytic reaction was initiated by the addition of DTT (3mM), BSA (0.5mg/ml), Evagreen (E.G.) (2.4x), dNTP (0.1mM), primer (1nM), Nb.BsmI (nickase) (400U/ml), BST polymerase (12.8 U/ml) to achieve the final concentration shown in brackets. The reaction was monitored by and the reaction monitored by evagreen fluorescence ($\lambda_{exc} = 460$ nm \pm 10 nm and $\lambda_{em} = 530 \pm 30$ nm). Negative fluorescence units are due to the background removal from control experiments containing proteinosomes and the reaction mixture in the absence of dNTP. The results show that the activity of encapsulated DNA is not affected by the encapsulation method. (Proteinosomes prepared by microfluidics shown in black and those prepared by bulk methods are shown in red). Negative RFU on the y axis due to background removals from control experiments which contain proteinosomes encapsulating DNA and reaction buffer in the absence of dNTPs.

Figure R3: Comparison of the encapsulation efficiency of (i) calibrated DNA into proteinosomes by (ii) microfluidic and (iii) bulk method preparations obtained by (i) calibration for (ii) microfluidics or (iii) bulk methods preparation. Total starting DNA concentration was 1 μ M. The results show that DNA encapsulation efficiency by microfluidics is lower and with smaller distribution 278 nm \pm 19 nm (RSD = 6%) and compared to bulk methods 370 nm \pm 88 nm (20%). Differences in encapsulation efficiencies in microfluidics (Supplementary figure 3 and 4) are attributed to experimental error that are most likely due to the variations in BS(PEG)₉ crosslinking efficiency.

Rebuttal Part 3: Calibration of DNA concentration within proteinosomes

Supplemental methods:

Determination of DNA concentration within proteinosomes after microfluidic encapsulation.

To determine the encapsulation efficiency of DNA-biotin-streptavidin complex within proteinosomes prepared by microfluidics a calibration curve for DY530- labelled DNA- biotin-streptavidin was undertaken using the Zeiss LSM 880 single point scanning confocal microscope using a 40x objective (C-Apochromat 40x NA 1.2W objective, Zeiss). To do this, a serial dilution of DY530-labeled DNA-biotin-streptavidin at concentrations of 0, 0.031, 0.063, 0.13, 0.25, 0.5 μM DNA were prepared in water and loaded into custom made channel slides. The samples were loaded onto the microscope and incubated at 25° C for 30 mins to ensure temperature equilibration. The microscope settings were consistent with those used for the autocatalytic and linear kinetic experiments (Figure 3) ($\lambda_{\text{DY530}}^{\text{exc}} = 514 \text{ nm}$ $\lambda_{\text{DY530}}^{\text{emi}} = 543\text{-}695 \text{ nm}$) and for the proteinosomes prepared for the effect of encapsulation on the autocatalytic reaction respectively.

The images were analysed using a FIJI macro code and the average fluorescence intensity at known DY530-DNA-biotin-streptavidin concentration was obtained and plotted. The data was fit to a linear regression equation to obtain the following equation Fluorescence (a.u.) = 4054.4 x (μM) +13.13, ($R^2=0.9975$) and used to convert fluorescence intensity within proteinosomes into concentrations. A Gaussian fit was applied to the histogram of number of proteinosomes vs concentration to obtain the average concentration and a standard deviation from 94 proteinosomes. 0.12 μM with a std=0.017 μM).

Supplementary Figure 6: Calibration of DNA concentration for proteinosomes produced by microfluidics. (Left) Serial dilution of DY530-labelled DNA-biotin streptavidin under the same microscope settings as the autocatalytic and linear kinetic experiments with proteinosomes. **(right)** Histogram showing the proteinosomes produced for reactions shown in Figure 3 and 4. Fluorescence units were converted to concentration using the calibration curve and its linear regression and a gaussian fit to the histogram of number of proteinosomes was undertaken to obtain the average concentration 0.12 μM +/-0.02 μM .

Amendments to existing text and figure captions:

Amendments to existing text and figure captions:

Figure 2: PEN DNA reactions are active within the proteinosomes (a) autocatalytic reaction, when $s = s'$ the PEN DNA reaction is autocatalytic (b) (i) Widefield optical fluorescence microscopy images showing increasing EvaGreen fluorescence from DNA production within proteinosomes. Final total concentration of DNA template is 0.06 nM , initial concentration of primer/substrate is 1 nM at 42°C . Scale bar is $20 \mu\text{m}$ (c) Autocatalytic growth curves obtained from image analysis show fluorescence increase of individual proteinosomes.

Figure 3: Characterisation of autocatalytic PEN DNA reactions within proteinosomes. (a) (i) with an increasing number of proteinosomes and therefore increasing overall template concentration and fixed exonuclease concentration. (a) (ii) Fluorescence widefield microscopy images, at $t = 0 \text{ min}$ and 160 min , showing proteinosome populations at different densities resulting in $0.06 - 0.52 \text{ nM}$ total DNA template. Scale bar is $100 \mu\text{m}$. (iii) Autocatalytic kinetic curves show the production of DNA inside proteinosomes, $0.06 - 0.52 \text{ nM}$ total DNA template at the population densities in (ii). Proteinosomes containing $0.12 \mu\text{M}$ template were incubated at 42°C in the reaction buffer containing 0.6 nM exonuclease and triggered with 1 nM of the substrate strand. (b) Reaction rates for autocatalytic reactions carried out with varying template concentration in (i) proteinosomes (blue squares) and (ii) buffer (orange circles). The reactions were triggered with 1 nM primer and incubated at 42°C in a well-plate reader or imaged in a microscope to obtain the reaction kinetics. The initial rates were obtained from fitting and plot as a function of template concentration. (c) (i) A linear fit of reaction rates for autocatalysis within proteinosomes at varying concentrations of total DNA template gives $392 \pm 104 \text{ min}^{-1} \cdot \text{mM}^{-1}$. Inset shows the residuals from the linear fit. (ii) Bar chart shows the linear fit of autocatalytic reaction rate per template (dr/dT) for (i) proteinosomes ($392 \pm 104 \text{ min}^{-1} \cdot \text{mM}^{-1}$) (blue), (ii) buffer ($2.3 \pm 1.1 \text{ min}^{-1} \cdot \text{mM}^{-1}$) (red). (d) (i) with constant template concentration and increasing exonuclease concentration, the exonuclease degrades the substrate inside and outside of the proteinosomes. With high exonuclease concentration, the reaction will be completely switched off. (ii) Initial rates of autocatalytic reaction with increasing exonuclease

concentration in proteinosomes (blue squares) and a buffer solution (red circles). Data was obtained from at least three repeats for at least three different experiments. The standard deviations (b-d) were obtained from the fit to the data.

Figure 4: Two-node networks based on encapsulated PEN DNA reactions within proteinosomes. (a) The product of a linear reaction programmed into population 1 activates autocatalysis in population 2. **(b)** Two different populations of proteinosomes are produced by encapsulating two different DNA templates; population 1 contains T_1 labelled with ROX and population 2 contains T_2 labelled with DY530. **(c)** (i) Dual-channel widefield microscopy show two distinct populations of proteinosomes. Population 1 (blue) containing DNA template T_1 labelled with ROX and population 2 (red) containing DNA template T_2 labelled with DY530. **(ii)** Time-lapse widefield fluorescence microscopy images show the increase in fluorescence intensity from non-sequence-specific intercalation of Evagreen Dye ($\lambda_{ex} = 475 \pm 28$ nm and $\lambda_{em} = 525 \pm 45$ nm) into the dynamically formed double-stranded DNA due to DNA production inside the proteinosomes for population 1 (blue arrow) and population 2 (red arrow), Scale bar is 20 μm . **(d)** Kinetic fluorescence profiles from individual proteinosomes from population 1 (blue curve) and population 2 (red) for a population ratio of 0.5:1 yielding template T_1 and T_2 concentrations of 0.035 nM and 0.068 nM, respectively. **(e)** Control experiment for the two-node network population. The experiment contained population 2 proteinosomes (final concentration of T_2 was 0.068 nM) and 50 nM of the S_1 primer and no proteinosomes containing T_1 . The plot shows the fluorescence intensity of EvaGreen as a function of time obtained from optical microscopy experiments. The red curve is from the proteinosomes containing T_2 and the grey curve is the fluorescence intensity from the background obtained from the area outside of the droplet. **(f)** Reaction kinetics of a two-node network cell population. Box plot showing the initial rates of the autocatalytic reaction (population 2) as a function of template concentration. Data points in the boxplot come from individual proteinosomes from different experiments 1:1 yielding template T_1 and T_2 concentrations of 0.067 nM and 0.068 nM, respectively. 0.5:1 yielding template T_1 and T_2 concentrations

of 0.035 nM and 0.068 nM, respectively and 0.25:1 yielding template T₁ and T₂ concentrations of 0.017 nM and 0.068 nM, respectively. The shaded colours in the plots in a-c are the standard deviations from the kinetic curves from individual proteinosomes in their corresponding populations. Data was from three different experiments from at least 11 proteinosomes.

The modifications to the main text are shown below:

From: “To this end, proteinosomes containing DNA template (1 μM) coding for an autocatalytic reaction,” **To:** “The DNA content within the proteinosome was determined by calibration (see supplementary materials and supplementary figure 6) to be 0.12 μM +/- 0.017 μM after the addition of 1 μM of DNA complex. “

From “As the concentration of template DNA within the proteinosome was fixed at 1 μM, the total concentration of DNA template in the dispersion was varied by altering the density of proteinosomes and was typically between 0.5-5 nM of DNA (Figure 3ai). The total concentration of template DNA in the dispersion was determined by measuring the total volume of proteinosomes within the capillary channel and obtaining the total concentration based on the assumption that each proteinosome contains 1 μM of DNA.” **To:** “As the concentration of template DNA within the proteinosome was fixed at 0.12 μM, the total concentration of DNA template in the dispersion was varied by altering the density of proteinosomes and was typically between 0.06 – 0.5 nM of DNA (Figure 3ai). The total concentration of template DNA in the dispersion was determined by measuring the total volume of proteinosomes within the capillary channel and obtaining the total concentration based on the assumption that each proteinosome contains 0.12 μM of DNA (supplementary figure 6 and 10).”

From: “The results show that in a dispersion of proteinosomes ([DNA template]_{total}= 0.5 nM)” **To:** The results show that in a dispersion of proteinosomes ([DNA template]_{total}= 0.06 nM)”

From: “high local concentration of template at 1 μM but a low total concentration (nM) in the total dispersion.” **To:** “high local concentration of template at 0.12 μM but a low total concentration (nM) in the total dispersion.”

From: “ The concentration of population 2 was kept constant whilst the concentration of population 1 was varied (Figure 4f, Supplementary figure 16) at T₁:T₂ ratios of 1:1 (0.56 nM : 0.57 nM) (Supplementary figure 16), 0.5:1 (0.29 nM : 0.57 nM) (Figure 4d), and 0.25:1 (0.14 nM : 0.57 nM)” **To :** The concentration of population 2 was kept constant whilst the concentration of population 1 was varied (Figure 4f, Supplementary figure 17) at T₁:T₂ ratios of 1:1 (0.067 nM : 0.068 nM) (Supplementary figure 17), 0.5:1 (0.035 nM : 0.067 nM) (Figure 4d), and 0.25:1 (0.017 nM : 0.068 nM) (Supplementary figure 17).

Methods:

From: “To do this, proteinosomes containing 1 μM of DNA template” **To:** “(Supplementary Figure 10). To do this, proteinosomes containing 0.12 μM of DNA”

From: “obtaining the total concentration based on the calibration that each proteinosome contains 1 μM of DNA” **To:** “obtaining the total concentration based on the calibration that each proteinosome contains 0.12 μM of DNA”

From: “1:1 (0.56 nM : 0.57 nM), 0.5:1 (0.29 nM : 0.57 nM), and 0.25:1 (0.14 nM : 0.57 nM).”

To: “of 1:1 (0.067 nM : 0.068 nM), 0.5:1 (0.035 nM : 0.068 nM), and 0.25:1 (0.017 nM : 0.068 nM).”

Supplementary figure 15. Autocatalytic kinetics in proteinosomes at different exonuclease concentrations. Proteinosomes containing 0.12 μM template were incubated in the presence of 1 nM DNA primer at 42 °C in a Spark 20 M well plate reader spectrophotometer (Tecan AG, Männedorf, Switzerland) using excitation and emission wavelengths of 470 ± 20 nm and 515 ± 15 nm respectively.

New Methods in the SI

Determination of DNA concentration within proteinosomes after microfluidic encapsulation.

To determine the encapsulation efficiency of DNA-biotin-streptavidin complex within proteinosomes prepared by microfluidics a calibration curve for DY530- labelled DNA- biotin-streptavidin was undertaken using the Zeiss LSM 880 single point scanning confocal microscope using a 40x objective (C-Apochromat 40x NA 1.2W objective, Zeiss). To do this, a serial dilution of DY530-labeled DNA-biotin-streptavidin at concentrations of 0, 0.031, 0.063, 0.13, 0.25, 0.5 μM DNA were prepared in water and loaded into custom made channel slides. The samples were loaded onto the microscope and incubated at 25° C for 30 mins to ensure temperature equilibration. The microscope settings were consistent with those used for the autocatalytic and linear kinetic experiments (Figure 3) ($\lambda_{\text{DY530}}^{\text{exc}} = 514$ nm $\lambda_{\text{DY530}}^{\text{emi}} = 543\text{-}695$ nm) and for the proteinosomes prepared for the effect of encapsulation on the autocatalytic reaction respectively.

The images were analysed using a FIJI macro code and the average fluorescence intensity at known DY530-DNA-biotin-streptavidin concentration was obtained and plotted. The data was fit to a linear regression equation to obtain the following equation Fluorescence (a.u.) = 4054.4 x (μM) +13.13, ($R^2=0.9975$) and used to convert fluorescence intensity within proteinosomes into concentrations. A Gaussian fit was applied to the histogram of number of proteinosomes vs concentration to obtain the average concentration and a standard deviation from 94 proteinosomes. 0.12 μM with a std=0.017 μM).

Supplementary Figure 6: Calibration of DNA concentration for proteinosomes produced by microfluidics. (Left) Serial dilution of DY530-labelled DNA-biotin streptavidin under the same microscope settings as the autocatalytic and linear kinetic experiments with proteinosomes. **(right)** Histogram showing the proteinosomes produced for reactions shown in Figure 3 and 4. Fluorescence units were converted to concentration using the calibration curve and its linear regression and a gaussian fit to the histogram of number of proteinosomes was undertaken to obtain the average concentration 0.12 μM +/-0.02 μM .

In addition, there are some typos:

1. As mentioned in page 22, a calibration curve relating the number of proteinosomes to their total intensity of fluorescence was explained in supplementary Figure 17. However, supplementary Figure 17 could not be found in the supporting information part.
2. Full stop was gone at the 14 line in page 10.
3. In supplementary Figure 11, there were several punctuations and word lost such as full stop and that "Polymerase" should be followed after "12.8 units.mL -1"

We thank the reviewer for these typos. They have been amended and the references to the supplementary figure 17 has been checked.

REVIEWERS' COMMENTS

Reviewer #1 (Remarks to the Author):

The authors have adressed all the concerns. The manuscript can be accepted in its current form.

Reviewer #2 (Remarks to the Author):

The revised manuscript has been improved, and most of the concerns raised by reviewers have been addressed effectively. Now, I have no further comments on the manuscript and support it to be published.